# Climatology of intermediate descending layers (or 150-km echoes) over the equatorial and low latitude regions of Brazil during the deep solar minimum of 2009

Ângela Machado dos Santos[1], Inez Staciarini Batista[1], Mangalathayil Ali Abdu[1,2], José Humberto Andrade Sobral[1], Jonas Rodrigues de Souza[1], Christiano Garnett Marques Brum[3]

[1]National Institute for Space Research (INPE) S. J. Campos, SP, Brazil
[2] Instituto Tecnológico de Aeronáutica, DCTA, São José dos Campos, Brazil
[3] Arecibo Observatory, University of Central Florida, Arecibo, Puerto Rico

*Correspondence to*: Ângela M. Santos (angela.santos@inpe.br, angelasantos_1@yahoo.com.br)

## Abstract

In this work, we have performed a study for the first time on the climatology of the intermediate descending layers (ILs) over Brazilian equatorial and low latitudes regions during the extreme solar minimum period of 2009. The result of this study shows that the occurrence frequency of the ILs is very high, being >60% over São Luís (2° S; 44° W, I: -5.7°) and >90% in Cachoeira Paulista (22.42° S; 45° W, I: -34.4°). In most cases the ILs occur during the day at altitudes varying from 130 to 180 km and they may descend to lower altitudes (~100 km) in a time interval of a few minutes to hours. The main driving force for the ILs at the low latitude region, may be considered to be the diurnal tide (24 h) followed in smaller dominance by the semidiurnal (12h), terdiurnal (8h) and quarter-diurnal (6h) components. In the magnetic equatorial sector, similar behavior was seen, with exception of semidiurnal tide, which in general does not appear to have influenced the IL's dynamics (except in summer). Additionlly, the ILs mean descent velocities over São Luís and Cachoeira Paulista show a day-to-day variability that may be associated to a wave like perturbation with a periodicity of some days. Some peculiarities in the ILs dynamics were noted, such as the presence of the ILs during the night hours. Ascending and descending ILs appeared to have been formed from some connection with the ionospheric F layer. Quite often, these characteristics are observed in the presence of strong signatures of the gravity wave propagation as suggested by the F layer traces in the ionogram. The descending intermediate layer over Brazil appears to have been formed through a process of F1 layer base detachment. An interesting case study showed

that an ascending ILs, initially detected at ~ 130 km, reached the base of the F2 layer, due probably to the gravity wave propagation and/or the effect of a prompt penetration electric field.

## 1 Introduction

The first observations on the existence of intermediate layers were reported in 1930s (Schafer and Goodall, 1933; Appleton, 1933; Ratcliffe and White, 1933). It was observed that this ionization layer, which is located between the E and F1 layers, occurred regularly in the height region of around 130 - 150 km. Since then, some studies have been carried out on the behavior of intermediate descending layers (IL's) over different longitude sectors using radar observations (Kudeki and Fawcett, 1993; Chau and Kudeki, 2006; Kudeki et al., 1998; Tsunoda and Ecklund 2004) and Ionosonde data (Rodger et al., 1981; MacDougall, 1974; MacDougall, 1978; Wilkinson et al., 1992; Szuszczewicz et al., 1995). Balsley (1964) showed that the existence of these layers located at ~150 km of altitude exhibited a downward movement during the daytime and ascending movement at dusk, with the intensity of the layer varying on a time scale of 5 to 15 minutes. Shen et al. (1976), observed that the ILs over Arecibo lasted for several hours. They also mentioned that in the valley region, the peak electron density of the layer ranged from ~$3x10^2$ to $1x10^3$ cm$^{-3}$.

According to Fujitaka and Tohmatsu (1973), the S2 and S4 propagation mode of atmospheric tide could be the dominant cause of the intermediate layers at night over middle latitude. Over the equatorial region, the ILs can possibly be caused by gravity waves (Kudeki and Fawcett, 1993). However, some characteristics observed also suggested that the phase velocity along the line-of-sight must be controlled firstly by the large-scale electrodynamics effect (driven by tides) and secondarily by the gravity waves of short period.

Some particularities of the IL such as its seasonality and cause-effect relationships have been studied over many years. Mathews and Bekeny (1979), for example, investigated the role of tidal winds in their diurnal and semidurnal components in the formation of ILs and concluded that the tidal winds could play a key role in the generation of these layers. Using ionosonde data, MacDougall (1978)

observed that the periodicity of the intermediate layers in the ionograms appeared to be related to semidiurnal oscillations. Mathews (1998) reported on the important role of the diurnal and semidiurnal tides in the generation and descent of the intermediate layers over Arecibo. Tsunoda (1994) suggested that a gravity wave wind driven interchange instability could be a possible generation mechanism of the field aligned plasma irregularities responsible for the echoes received from the IL.

Regarding the study of the ILs using radar data, the first observations of the "150-km echoes" over Brazilian equatorial region were made by de Paula and Hyssel (2004). The radar echoes were identified at around 09:00 LT at ~ 165 km altitude, and by 12:00 LT the echoes descended to ~ 145 km. A gradual upward movement of the ILs reaching 160 km of altitude was observed in the following hours. At 14 LT the IL disappeared, presenting in this way a local time variation resembling a "necklace" shape. Similar pattern was observed in different longitudinal sectors. From the analysis of the Range-Time-Intensity (RTI) maps, Rodrigues et al. (2011) showed that the lowest rate of occurrence of the 150-km echoes in Brazil was during the March equinox, whereas the strongest and longest duration echoes were observed between June and September. Another important finding by Rodrigues et al. (2011) was that there was an apparent variability of the Doppler displacement with height, indicating that some of ILs over Brazil might have a different formation mechanisms from those operative in other longitudinal sectors. The ILs observed with the 30 MHz radar at São Luís presented a thickness of 3 to 5 km and were located between 140 and 170 km of altitude.

In this paper, we present for the first time, the climatology of the intermediate layers over the equatorial and low latitude locations in Brazil during a period of extremely low solar activity. The important points to be discussed in this paper include the influence of atmospheric tides and gravity waves in the ILs dynamics and the possible contribution of disturbance electric fields in some specific cases. Some peculiarities found in the ILs characteristics over Brazil will also be discussed.

## 2 Methodology and data presentation

The observational data analyzed in this work were obtained from the Digisondes operated at São Luís – SL (2° S; 44° W, I: -5.7°) and Cachoeira Paulista - CP (22.42° S; 45° W, I: -34.4°) during the

period of extreme solar minimum activity of 2009. The Digisonde is an ionospheric radar composed by a transreceiver system that emits pulses of electromagnetic energy at frequencies ranging from 1 to 30 MHz. The electromagnetic signal is transmitted vertically to the ionosphere, with peak power of the order of 10 kW in the case of the Digisonde DGS256 (over Cachoeira Paulista) and a peak power of the order of 300 W for Digissonde DPS-4 (over São Luís). The echoes of the signals  reflected by the ionospheric layers are registred as ionograms in the form of graphic plot of frequency versus virtual height. The Digissonde data is preprocessed through the ARTIST software (Automatic Real Time Ionogram Scaler with True Height) and later manually edited by the SAO-Explorer (Standard Archive Output Format) software. For more details about Digisonde, see for example Reisnish (1986) and Abdu et al. (2009b). The São Luis and Cachoeira Paulista  Digisonde data were acquired at a cadences of 10 and 15 min, respectively.  The data over SL was analyzed from March to December and that over CP from January to December. The local times at the two sites are: LT = UT- 3h. The following criteria were established in order to ubiquitously classify a certain layer as intermediate layer:

(a) When the critical frequency of the lower layer (extraordinary trace) exceeds the minimum frequency of the ordinary trace of F layer, the layer in question was considered as IL. Otherwise, the layer was classified as a regular E2 layer. Depending on the height of this layer, we carefully evaluated the identification of the IL case by case;

(b) In the cases in which the characteristics described in item (a) were not clear, a sequence of ionograms was used to classify the type of the layer;

(c) When a new layer appeared to be formed from a detachment of F1 layer, the IL was only considered when such detachment was total;

(d) For a descending layer to be classified as IL it should first occur at or above 130 km;

(e) When the studied layer was initially detected below 130 km, but in the sequence of the ionograms an ascending movement reaching h ≥ 130 km was observed, the layer was classified as an IL;

**(f)** The ILs height and frequency parameters were extracted from the extraordinary trace. Besides that, they were processed during their downward movement until they merged into sporadic-E (Es) layers.

It is important to note that, according to the criteria mentioned above, the layers considered here as "intermediate" may have evolved into a normal c-type sporadic-E layer, for example. Figure 1 shows an example of an intermediate layer located at ~ 150 km (h'IL) with a top frequency (ftIL) of 4.51 MHz. We can see that the minimum frequency of the F layer (~ 4.1 MHz) was lower than the top frequency of IL, thus satisfying the criterion described in item (a).

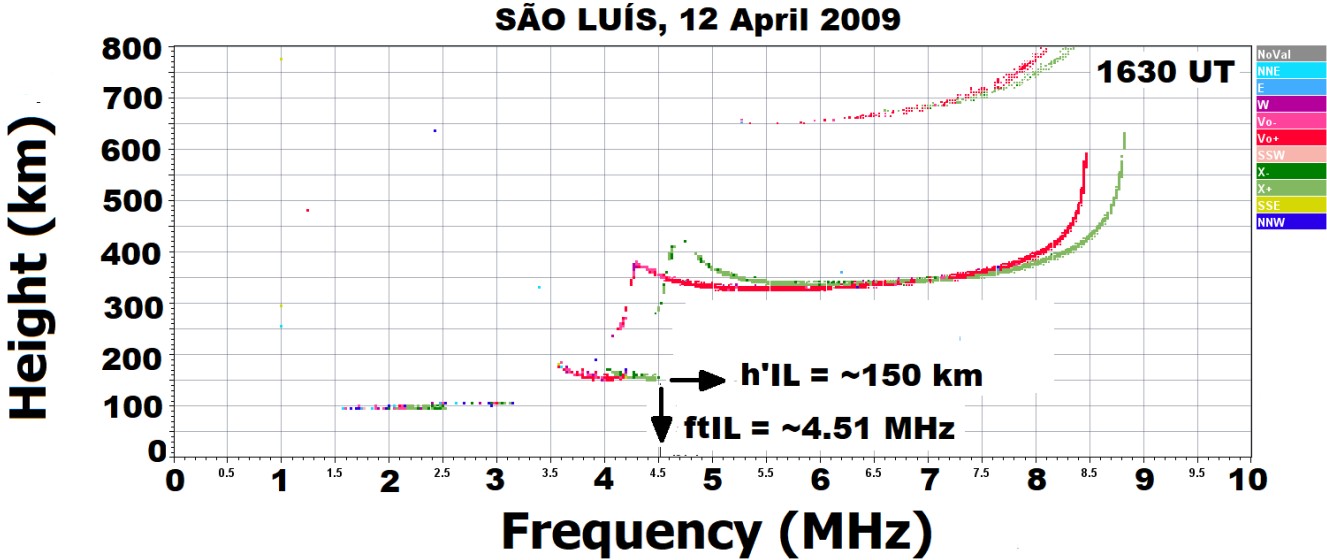

**FIGURE 1**: Ionogram over São Luis on 12 April 2009. The virtual height (h'IL) and the top frequency (ftIL) of intermediate layer are indicated in this Figure. The ordinary and extraordinary traces are represented by red and green colors, respectively.

## 3 Results and Discussion

Figure 2 presents the month-by-month variation of the percentage of occurrence of the intermediate layers over SL and CP. This analysis was performed based on the informations given in Table 1. The percentage occurrence was calculated dividing the number of the days in which ascending or descending

intermediate layers were detected (regardless whether they were observed more than once during a day or not) by the number of days of available data. The upper panel of Fig. 2 shows the statistics over SL. It is interesting to observe the high occurrence rate (above 60%) during 2009 over this station. In July and August, the monthly occurrence reached 100%. Over Cachoeira Paulista, the occurrence was even more pronounced throughout the period analyzed, reaching 100% in the months of April, June, July and December.

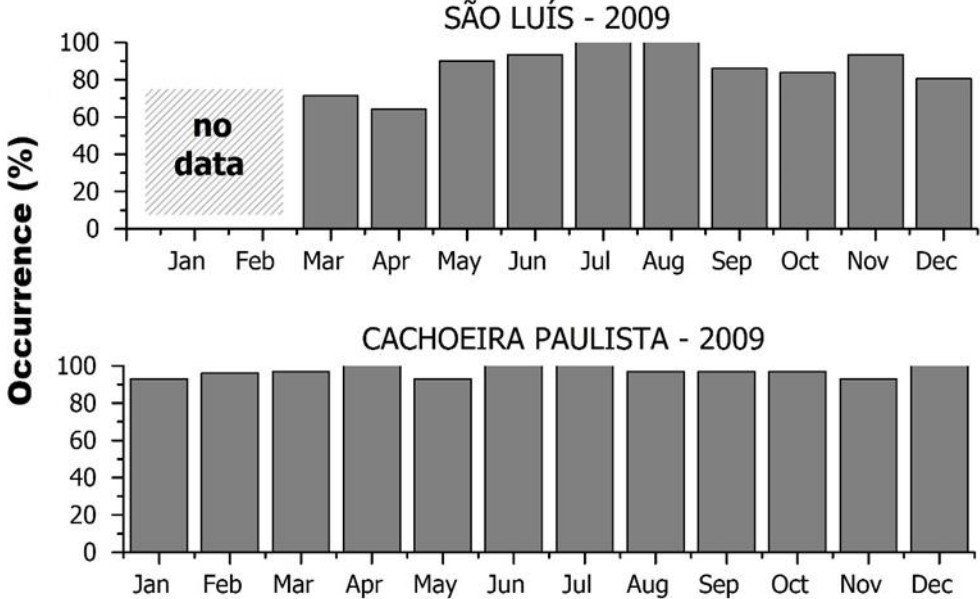

**FIGURE 2**: Monthly percentage occurrence of the descending intermediate layers for solar minimum period of 2009 at São Luís (upper panel) and Cachoeira Paulista (bottom panel). In upper panel the lack of data during January and February is marked.

Table 1 - Number of days used in our study.

| SÃO LUÍS | Jan | Feb | Mar | Apr | May | Jun | Jul | Aug | Sep | Oct | Nov | Dec |
|---|---|---|---|---|---|---|---|---|---|---|---|---|
| Number of days with data | -- | -- | 21 | 28 | 31 | 30 | 31 | 30 | 30 | 31 | 30 | 31 |
| Number of days with ILs | -- | -- | 15 | 18 | 28 | 28 | 31 | 30 | 26 | 26 | 28 | 25 |
| | | | | | | | | | | | | |
| CACHOEIRA PAULISTA | Jan | Feb | Mar | Apr | May | Jun | Jul | Aug | Sep | Oct | Nov | Dec |
| Number of days with data | 31 | 29 | 28 | 30 | 31 | 30 | 31 | 31 | 30 | 31 | 28 | 31 |
| Number of days with ILs | 29 | 28 | 27 | 30 | 29 | 30 | 31 | 30 | 29 | 30 | 26 | 31 |

## 3.1 Seasonal and diurnal variations

Figure 3 shows mass plots of the virtural height (h'IL) and top frequency (ftIL) of the intermediate layers as a function of time. The data sets were grouped into equinoxes (March-April; September-October), winter and summer solstices (May-to-August and November-to-February, respectively) seasons. Note that the data availability and occurrence of the IL's for these periods are summarized in Table 1. The main results of this analysis for SL and CP can be summarized as follows: **(a)** the IL is a phenomenon that occurs predominantly during daytime; **(b)** in general the higher top frequencies are observed for the intermediate layers that descend in height starting at maximum height of 130 km; **(c)** Over São Luís, nocturnal intermediate layers were observed between 01:00 and 08:00 UT (22:00 and 05:00 LT) in the equinoxes and summer solstice, but not in the winter during the same period. A few cases were observed also after 21:00 UT. Over CP, the nocturnal ILs were observed between ~ 22:00 and 09:00 UT during the three seasons; **(d)** At dawn (~ 06:00 LT), the lowest average top frequency of the intermediate layers over SL was ~ 2.5 MHz during equinoxes and in summer, and ~ 2.1 MHz in winter solstice. In CP, these values were ~ 2.5 MHz in equinoxes, 2.1 MHz in summer and 1.9 MHz in winter; **(e)** Ascending and descending layers were observed at altitudes above 200 km during the equinoxes and winter over SL, but it is important to observe that the summer months are underrepresented due to the absence of data for January, February and some days in March. Over CP, it is possible to identify layers at altitudes slightly higher than 200 km during the three seasons. In all cases, during the interval between 05:00 and 07:00 LT (08:00-10:00 UT), the classification of the layers as ILs turned out to be difficult because this is the time during which the ionospheric E layer begins to be registered in the ionograms.

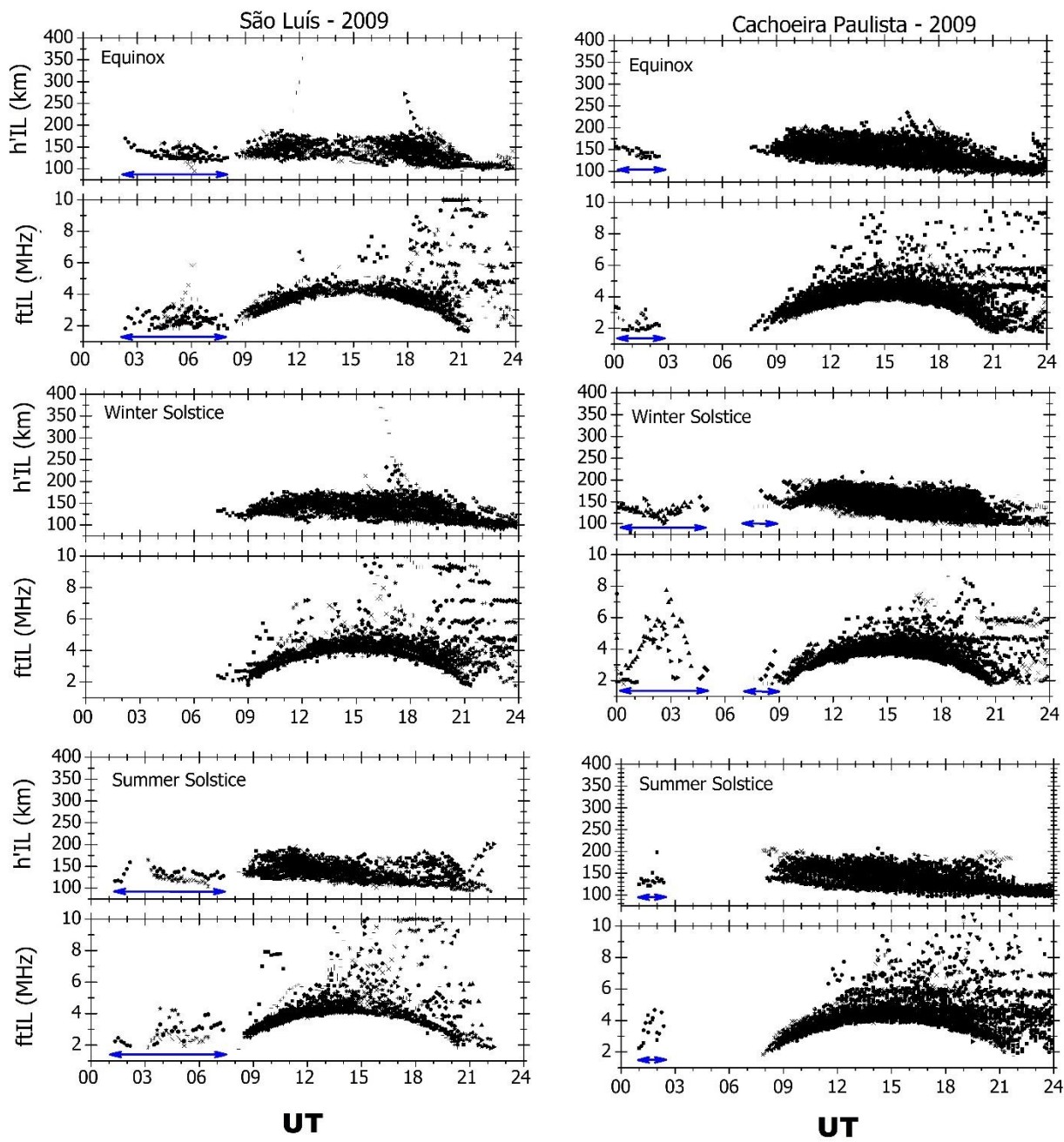

**FIGURE 3**: Time dependence of intermediate layer parameters over São Luís and Cachoeira Paulista for different seasons (column of panels of left and right, respectively). The horizontal blue arrow indicates the presence of nocturnal layers between 00:00 and 09:00 UT.

Figures 4a and 4b show the diurnal variation of the ILs parameters over SL and CP, respectively, mass-plotted for each day of the month. It is interesting to observe that in many cases, the layer descended to heights less than 130 km after ~ 15:00 UT and 18:00 UT. In October, November and December, the layer descent started earlier, that is, soon after ~09:00 UT and 13:00 UT. The ILs were not observed in March and April between 12:00 and 15:00 UT. Over CP (Figure 4b) it is possible to note the ILs tendency to reach altitudes below 130 km some hours earlier when compared with that over SL, except in November and December. In addition, in almost all the cases, the ILs were formed starting 09 UT on, at both SL and CP. For a better visualization, the maximum limit of the vertical axes of Figures 4a and 4b was fixed at 200 km, but as can be seen in Figure 3, on some days, the ILs were registered at altitudes well above this limit.

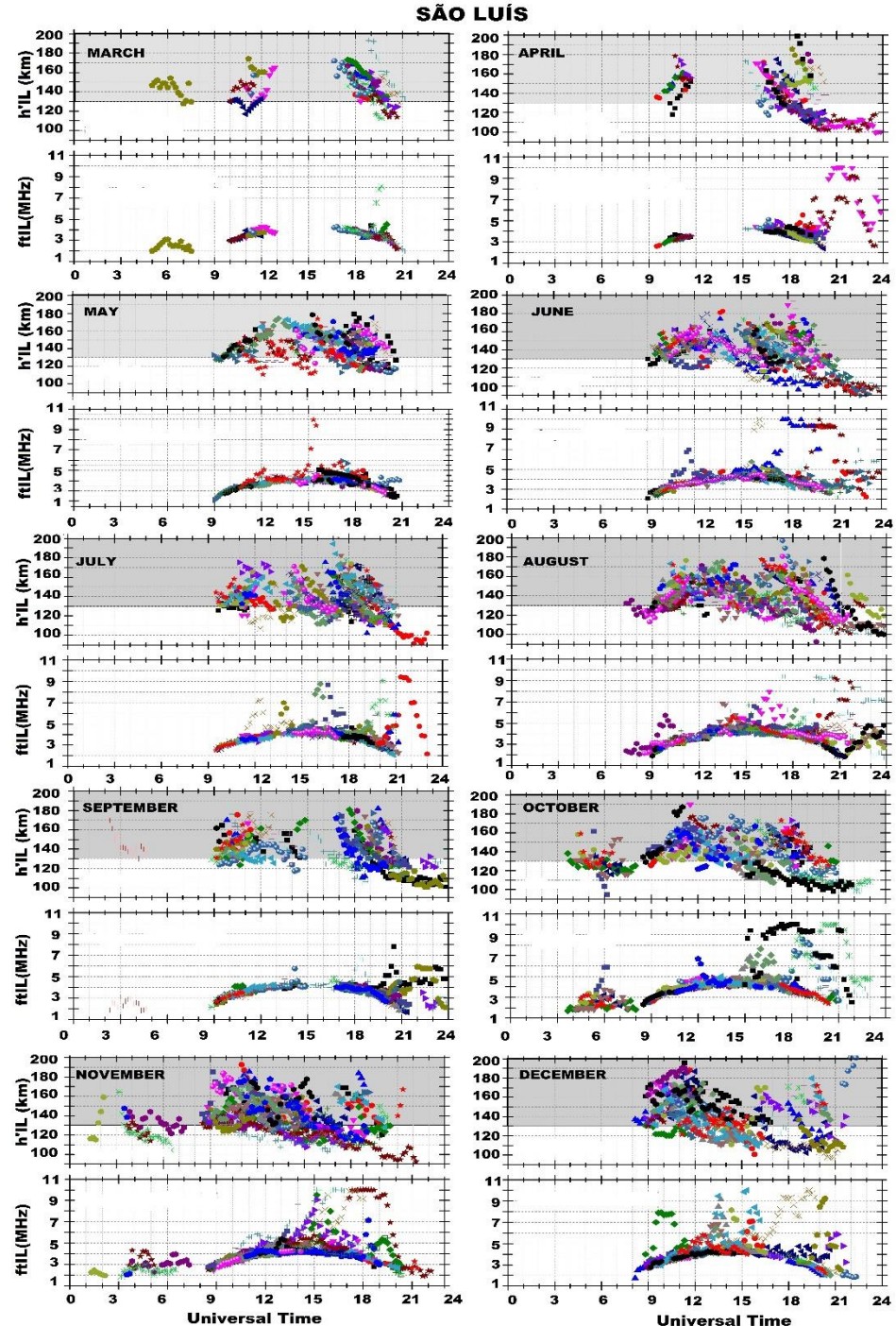

**FIGURE 4a**: Time variation of the virtual height (h'IL) and top frequency of intermediate layers (ftIL) over São Luís from March to December 2009. Different colors are used to represent the different days of each month.

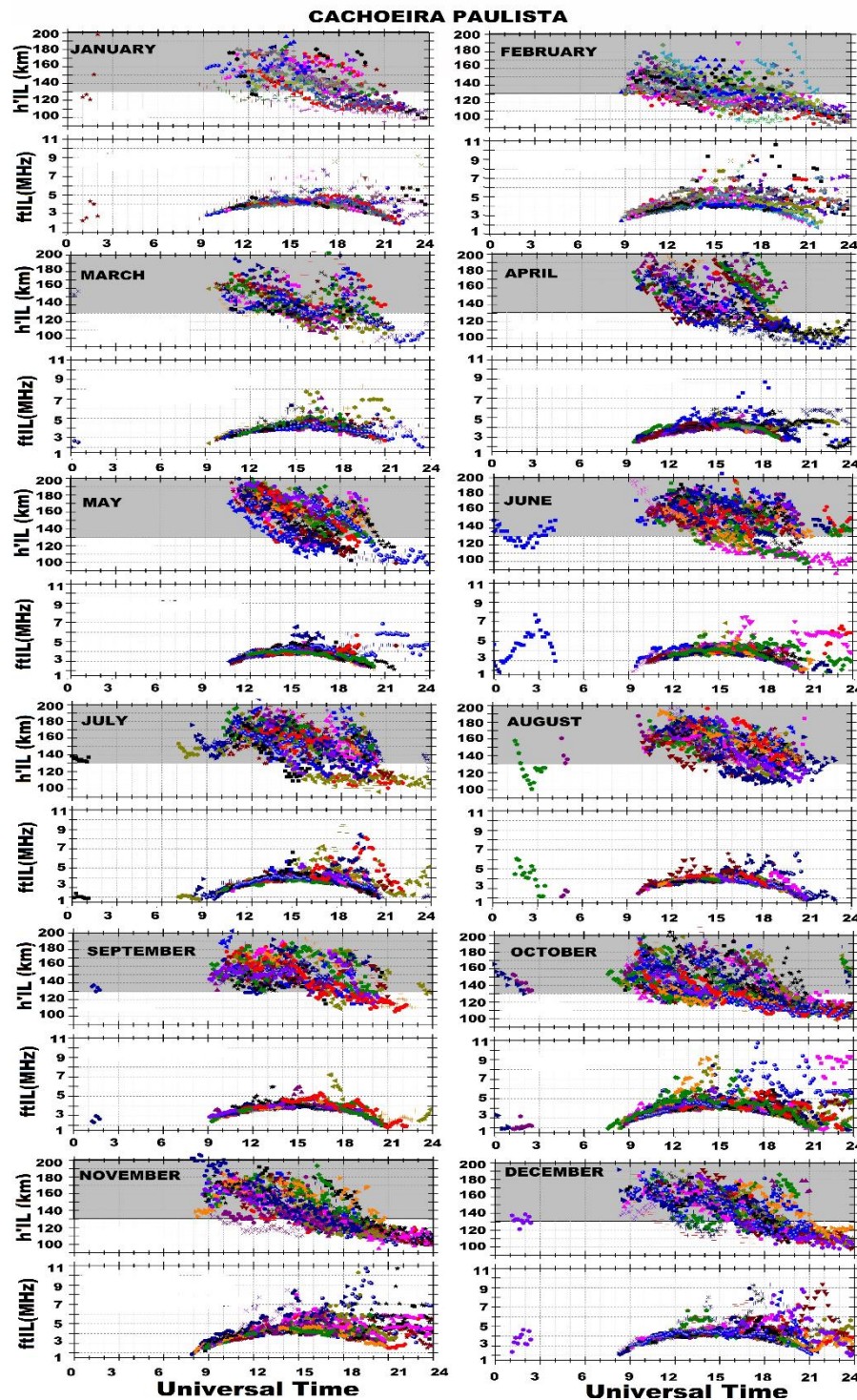

**FIGURE 4b**: Same as Figure 4a, but for Cachoeira Paulista from January to December 2009.

In general, the results presented in Figure 4 make it clear that the height at which the ILs were detected is quite variable and the behavior of the layer top frequency is very similar to that of the E-layer critical frequency with a maximum around local noon (15:00 UT). In most of the cases, the ILs present descending movements, but on some days, an ascending movement was also identified. Besides that in our data base we verified two types of ILs occurring around the same time; the ILs that may or may not descend to altitudes less than or equal to 130 km; the ILs that appear and disappear several times a day and the ILs that may be connected in some way with the F layer (that is, at h $\geq$150 km). More details about these specific features will be given in the next sections.

The analysis of the Digisonde data presented here reveals, for the first time, a higher occurrence rate of the ILs during the solar minimum period with some distinct features. Previous studies have suggested that the ILs could result primarily from the wind shear processes driven by different tidal modes (Fijitaka and Tohmatsu, 1973; Mathews and Bekeny, 1979; Tong et al., 1988). Using E-region the MIRE model, Resende et al. (2017a, 2017b) verified that the diurnal component of the zonal wind is very important for the formation of the Es-layers (at 90-140 km) over Brazil; however, the descending movement is simulated only when the meridional wind is included in the model. Lee et al. (2003) reported that, at a Japanese location over the equatorial ionization anomaly, the semidiurnal tidal mode was dominant during spring and winter, while the quarterdiurnal mode prevailed in summer/autumn. For an Indian station, Niranjan et al. (2010) observed a high occurrence probability of the ILs during winter, moderate occurrence rate during equinox and low occurrence during summer solstice.

As presented in Fig. 3, the occurrence of ILs over the equatorial and low latitudes regions does not present any seasonal preference, but the daily occurrence probability shows an interesting perspective. Figure 5 shows the seasonally averaged local time variation of the occurrence of ILs events (in percentage) for the sites in discussion. We note that during the equinoxes, the ILs occurrence above 130 km over SL shows two maxima, one at 11:00 UT and other at 18:00 UT, each one with occurrence probability of ~ 25-30%. Very small occurrence probability was observed during the night (~ 5%). Further, the occurrence of the descending layers predominantly during the daytime hours in Figure 5 appears to suggest the possibility of an important role of the E-layer dynamo on the dynamics of the descending layer. A similar behavior is observed in the winter solstice with an occurrence rate of ~ 35% for both peaks. In this case, the nocturnal ILs were not observed. In the summer months, only one dominant peak was observed at 12:00 UT with an occurrence probability of 60%. During the night, the occurrence was below 5%. Unlike over São Luís, the maximum probability of IL occurrence over Cachoeira Paulista was characterized by two broad peaks at around 12:00 UT and 15:00 UT for all periods analyzed with the probability of occurrence variyng between 50% and 70%. It is important to mention that the calculation of the occurrence probability in Figure 5 took into consideration all the simultaneous intermediate layers' events. For this reason, there are some differences in the number of occurrence used in this analisys (indicated in each panel of Figure 5) and those used in Figure 2 (based in values of Table 1) because some days can have more than one IL event.

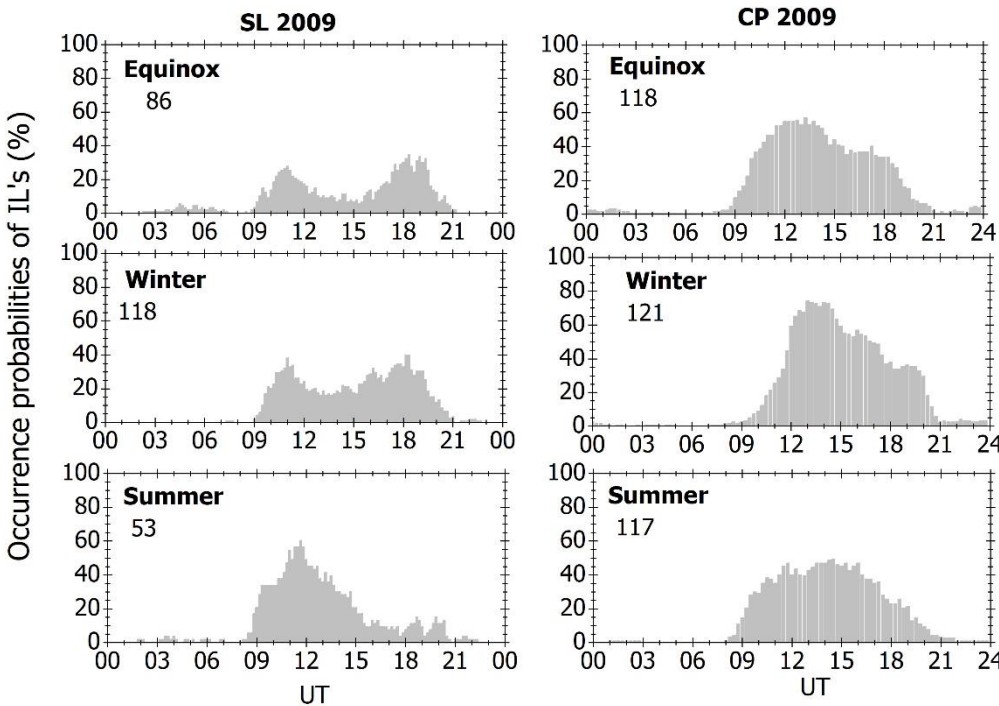

**FIGURE 5:** Occurrence probability of the intermediate layers (h≥130 km) events over SL and CP during 2009 (column of panels of left and right, respectively). In this plot, all the simultaneous ILs observed were considered as distinct events. The number of occurrence of IL events is indicated in each panel.

It has been known from previous studies that the formation of the intermediate layers are influenced by atmospheric tides, mainly by the semidiurnal mode. In order to investigate how these waves can affect the ILs dynamics over the Brazilian equatorial and low latitudes regions, we applied the Fast Fourier Transform (FFT) analysis on the IL height parameter and found that the semidiurnal mode could influence the ILs mainly over Cachoeira Paulista. However, the dominant influence in all the cases, comes from the diurnal tide for both SL and CP. In the present analysis, the data gaps, which indicate the non-occurrence of the IL (with exception in January and February for SL) were replaced by the number "zero".

Figure 6 presents the result of the FFT analysis of the IL heights for São Luis (top panel) and for Cachoeira Paulista  (bottom panel). Over São Luis, the equatorial region, the descending ILs present a well-defined diurnal periodicity in all seasons, followed by terdiurnal and quarterdiurnal modes. There is an exception in summer, when the contribution of semidiurnal mode is also evident. However, as mentioned previously, the absence of data during January and February could affect this result. Similar to the observation over SL, the role of the diurnal tide is dominant over CP (see lower panel of Figure 6), but the influence of semidiurnal mode was evident, mainly in the equinoxes and winter. A little influence from the terdiurnal and quarterdiurnal modes can also be identified during the equinoxes and winter. As mentioned by the Niranjan et al. (2010), the ILs formation is primarily controled by the shear forces driven by meridional and zonal winds, and their day-to-day variability (as evidenced in Figures 4a and 4b), can be attributed to variations in the atmospheric tides, gravity waves, electric fields and metallic ions populations as well.

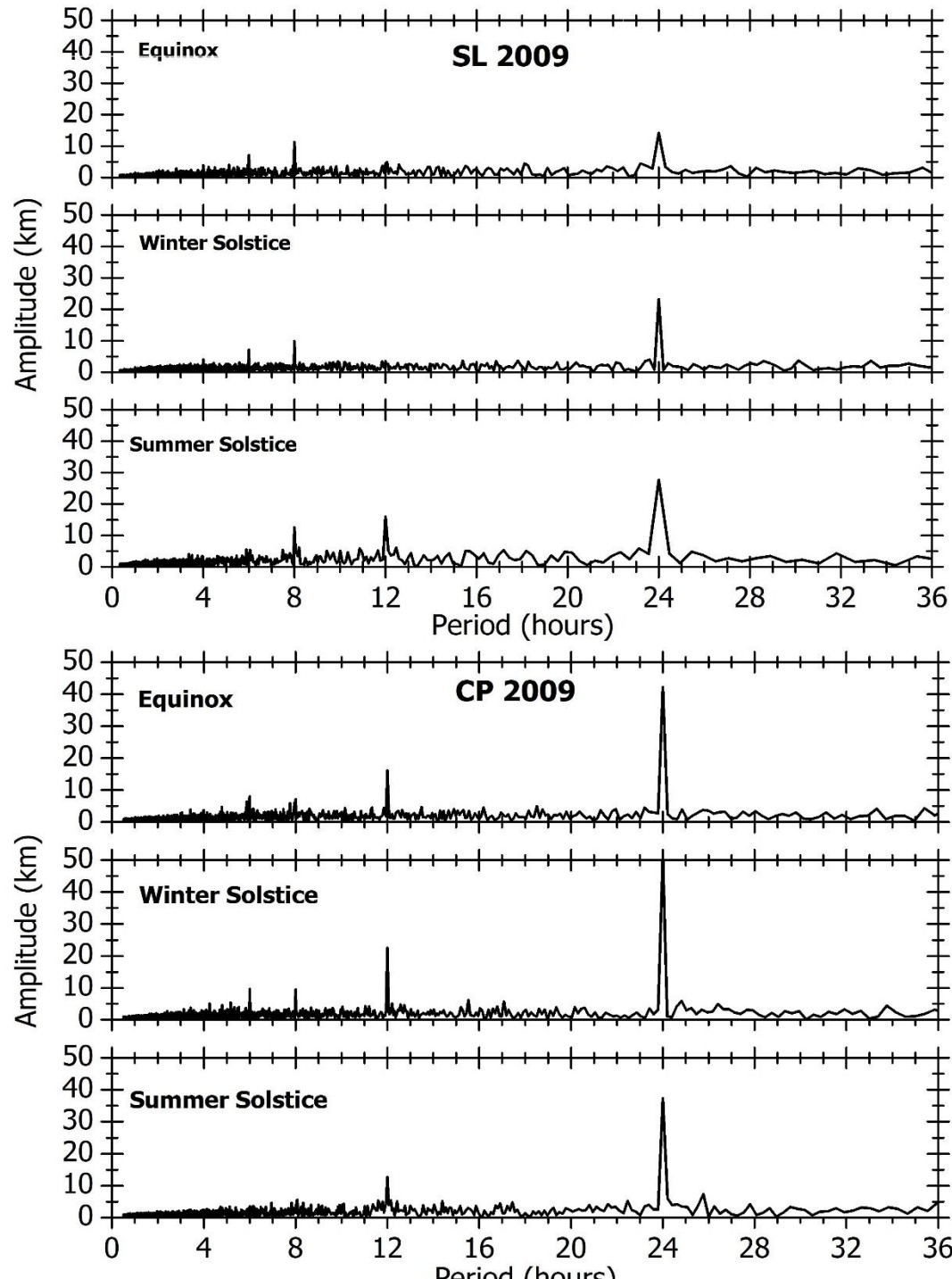

**FIGURE 6:** Period amplitude of the intermediate layer heights obtained from the FFT analysis for SL (top) and CP (bottom) during 2009.

The results of Figure 6 reveal that the influence of atmospheric tides in the formation/development of intermediate layers can have a latitudinal variation, being higher over Cachoeira Paulista than over São Luis. This behavior is in accordance with Andrioli et al. (2009a), who showed, for example, that the diurnal tidal amplitude over the low latitude regions of Cachoeira Paulista and Santa Maria (SM, 29.7°S; 53.8°W) is higher when compared to the amplitudes over the equatorial site, São João do Cariri (7.4°S, 36.5°W) (or São Luis). Whilst the zonal component of the wind presented a semiannual variation over São João do Cariri (maximum at the equinox) and annual and semiannual variation over CP and SM, the meridional component presented semiannual pattern over the three locations studied. Resende et al. (2017a, 2017b) also reported that the amplitudes of the meridional and zonal winds over Cachoeira Paulista are higher than over São João do Cariri. More information about the variability of meridional and zonal winds and their diurnal and semidiurnal oscillations over Brazilian regions can be found in Batista et al. (2004), Lima et al. (2007), Buriti et al (2008) and Andrioli et al. (2009b).

**3.2 Nocturnal intermediate layers**

The occurrence of nocturnal intermediate layers is one of the peculiarities found in our studies. Over São Luís, the highest number of cases was detected between October and November. Figure 7a shows some examples of the nocturnal ILs occurring between 0120 UT (2220 LT) and 0830 UT (0530 LT) over SL. The times in the ionograms do not necessarily indicate the exact moment in which the intermediate layer was formed, but they indicate the times at which they could be better visualized. It is

interesting to note that in these examples, the nocturnal layers exhibited a similar shape in almost all cases, with a straight and "spreading" base appearance. The height/top frequency of these layers varied between ~ 130 and 170 km / 2.2 and 3.5 MHz. Similar to the diurnal intermediate layers, the nocturnal layers also had descending and ascending characteristics.

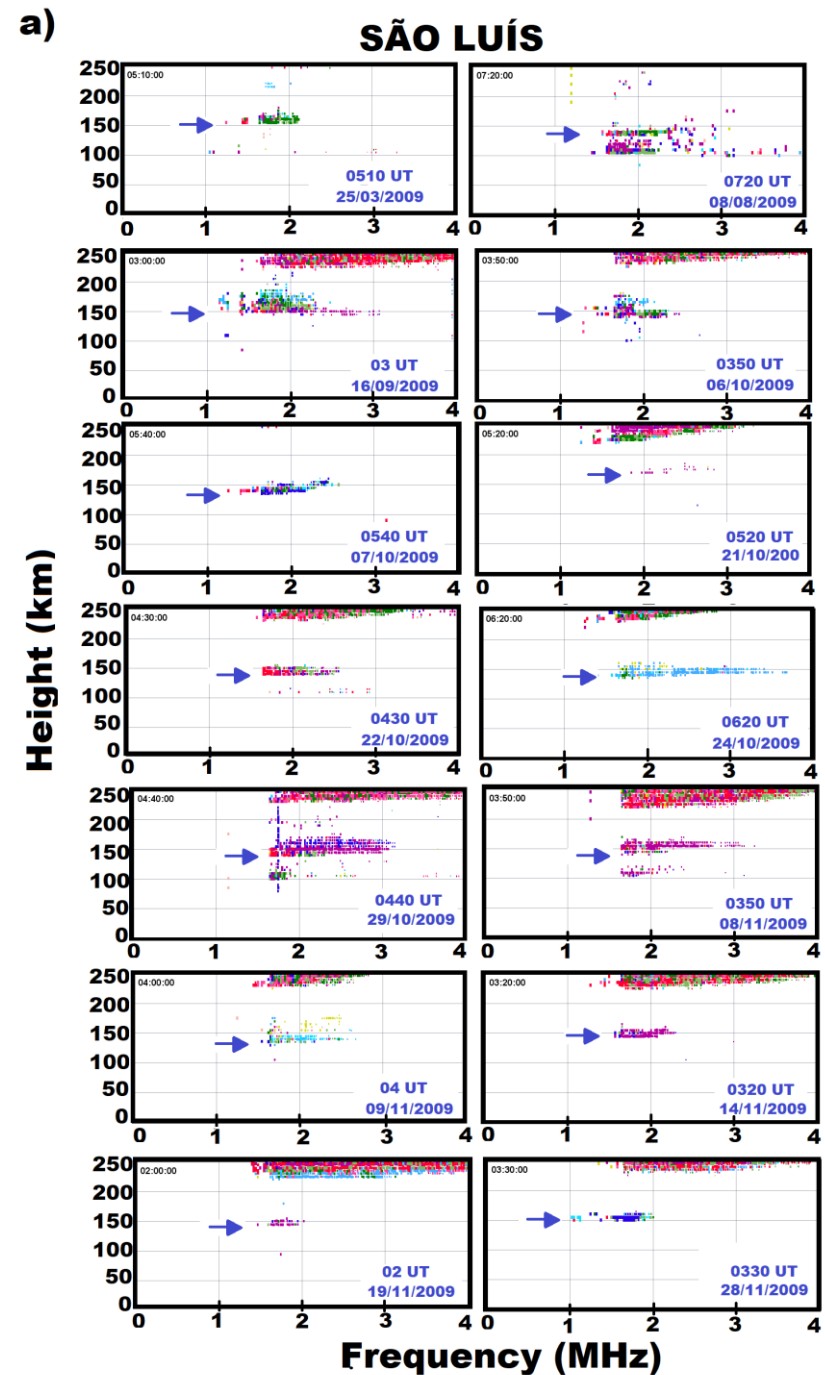

**FIGURE 7**: a) Ionograms over São Luís showing some examples of nocturnal intermediate layers on different days.

Over Cachoeira Paulista also the nocturnal intermediate layers were registered. The highest occurrence turned out to be in June and October (5 days in both months). The panels of Figure 7b show an example of this type of layer along the night of August 10. At around 01:30 UT an IL is observed at ~150 km. The subsequent sequence of ionograms shows that the IL rapidly descended to ~ 110 km. The range spreading characteristic was observed also in the nocturnal ILs, which in this case presented a range of ~ 50 km. On the other hand, a different shape was found for the IL on 20 November, starting at 0837 UT, as shown in the panels of Figure 7c. We can observe at 200 km a curved format like that of the 'h' type Es-layer. The sequence (not presented completely here) shows that this layer descended only a few kilometers and at 10:37 UT the layer was already connected to the F1 layer.

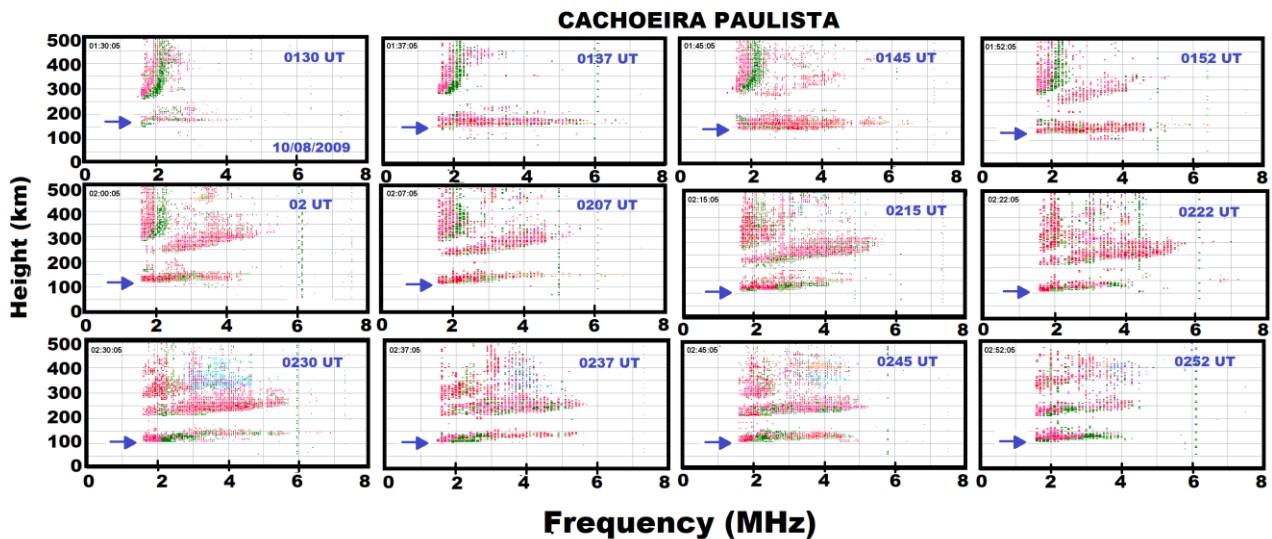

**FIGURE 7: b)** Same as Figure 7a, but for Cachoeira Paulista during August, 10.

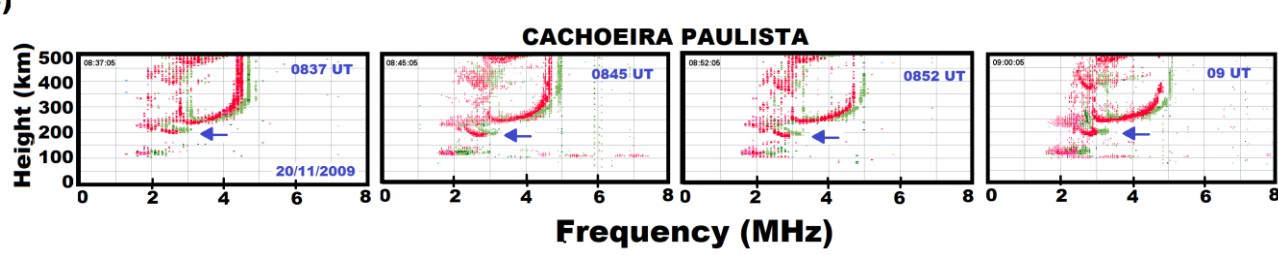

**FIGURE 7: c)** Nocturnal intermediate layer over CP starting with a retardation in height similar to that of the h-type sporadic layer.

As mentioned by Lee et al. (2003), the detection of nocturnal intermediate layers may be influenced by a limitation in the Digisonde, when the plasma frequency of the IL becomes lower than the lowest sounding frequency (1MHz), or when a blanketing is caused by an underlying Es layer (l/f type). Over Arecibo, a middle latitude location, for example, the nocturnal IL is a very common phenomenon (Shen et al. 1976; Osterman et al. 1994). Rodger et al. (1981) also reported the presence of nocturnal ILs over South Georgia. The authors mentioned that the behavior of the F-layer some time before the occurrence of the IL suggested that the increase of ionization observed in the region above ~130 km and in the lower F region resulted from a downward transport from F layer probably due to the solar semidiurnal tide. According to them, the probability of occurrence of the ILs is very low when the minimum virtual height of the F2 layer base is higher than 220 km, but increases rapidly as it falls towards lower altitudes. As shown in the examples of Figure 7a, in almost all the cases for SL, for example, we noted that the F layer was located at ~ 225 km, but there are some exceptions, like that on 25 March, 8 August and 7 October. In those cases, the F region was slightly above 250 km. Over CP, the nocturnal layers had similar characteristics. On 10 August, we can see clearly that the descent of the intermediate

layer occurred during an ascending of the F layer until 02:00 UT. It is interesting to note also the similarity of the IL during this day with the sporadic-E layer with high range spreading, probably formed due to particle precipitation (a-type Es layer).

 **3.3 Simultaneous ILs and those that are connected to the F region**

Besides the occurrence of simultaneous ILs, we identify some cases in which the formation of the ILs was connected to the F region. They are: a) intermediate layers formed at the high frequency end of the F1 layer (IL'F1t), b) ILs formed from a detachment of the F1 layer base (IL'F1b), c) ILs formed from a perturbation in the F2 layer base (IL'F2b), and d) ascending ILs that reached the base of F2 layer. Among the above cited, the most common case observed at both São Luís and Cachoeira Paulista was those in which the layers formed from a detachment of the F1 layer base.

Figure 8 shows an example of ascending intermediate layer over SL that was located initially at ~ 130 km. At 11:00 UT, this layer presented a weakening followed by an intensification after 10 minutes. Between 10:20 and 11:20 UT, we can observe an apparent ascending movement of the IL (from 132.9 to 153.5 km) and its subsequent merging with the F1 layer at 11:30 UT. From this time on, an extra stratification at the high frequency end of the F1 layer with an ascending structure was observed and attained the F2 layer base at 1230 UT (400 km). It is also interesting to note the changes that occurred in the F2 layer trace during this period.

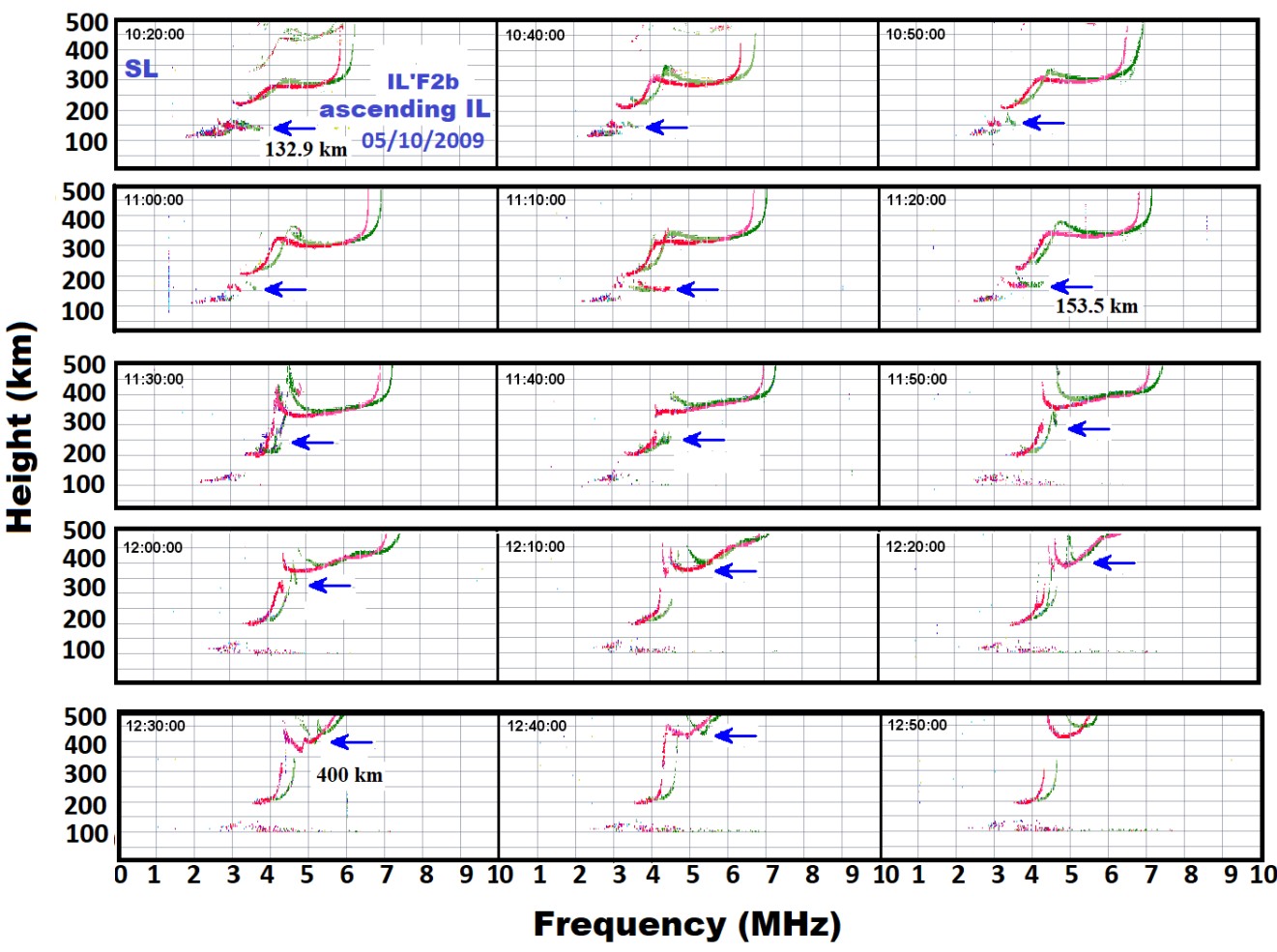

**FIGURE 8**: Sequence of ionograms taken during 1020 – 1250 UT on 05 October 2009 showing the presence of ascending intermediate layers over São Luís. The blue arrow indicates more clearly the raise of IL.

Using radar data, some authors have shown also ascending movement of the layer in the region of ~ 150 km (see for example de Paula et al. (2004), Patra et al. (2007) and Li et al. (2013)), but hardly any discussion about them has been made until now. In order to understand the cause of this layer rise, we show in Figure 9 some parameters of the F layer and IL during this day (05 October 2009). In the first panel are presented the F-layer virtual height (h'F), the F1 and F2 layer peak heights (hmF1, hmF2), and

the minimum virtual height of the IL trace (h'IL). The last two panels present the distribution of the tropospheric convective zones as obtained from the GOES satellite above Brazil at 07:00 and 08:00 UT. We may note that at ~11:10 UT, the IL started to rise from ~ 150 km reaching 420 km at an interval of one hour and a half. Around the same time, the hmF2 increased but in a smoothed way. Between 12:00 and 12:40 UT, the h'IL (virtual height) was heigher than hmF2 (real height). After the intermediate layer joins the F1 layer (see the ionogram at 11:30 UT in Fig. 8), the h'IL was considered as the height in which a perturbation was observed in the high frequency end of the F1 layer until it attained the base of F2 layer. The lower two panels show images of the tropospheric convective zones (with the temperatures below -60º C) around São Luis (at distances of ~700 and 1000 km) taken some hours before the increase in the height of IL (07:00 – 08:00 UT), which might provide some support to the idea about the role of gravity waves in the raise of the IL. According to Vadas and Fritts (2004), the upward propagating gravity waves excited by tropospheric convective systems could have strong impacts in the atmosphere at high altitudes due to the fact that they have long vertical wavelengths and propagate in all directions from the convection source. Besides that, some peculiarities in the ionograms during this day,in the form of the forking trace provide another evidence for the influence of the gravity waves in this event (see the last ionogram in Figure 13).  However, the increase in the AE index from 25 to 175 nT at 1130 UT, appears to have produced some effect through the action of a prompt penetration electric field (PPEF) with eastward polarity in the behavior of the IL height at this time too. It is interesting to observe that the F1 region peak height (hmF1) showed a steady increase between 1130 UT and 1215 UT while the hmF2 and h'F did not respond to this disturbance electric field. Although the hypothesis that a PPEF could have influenced the

IL movement during this event, the reason for the distinct responses of the ionospheric parameters to the penetration electric field is not completed understood and would need more investigation.

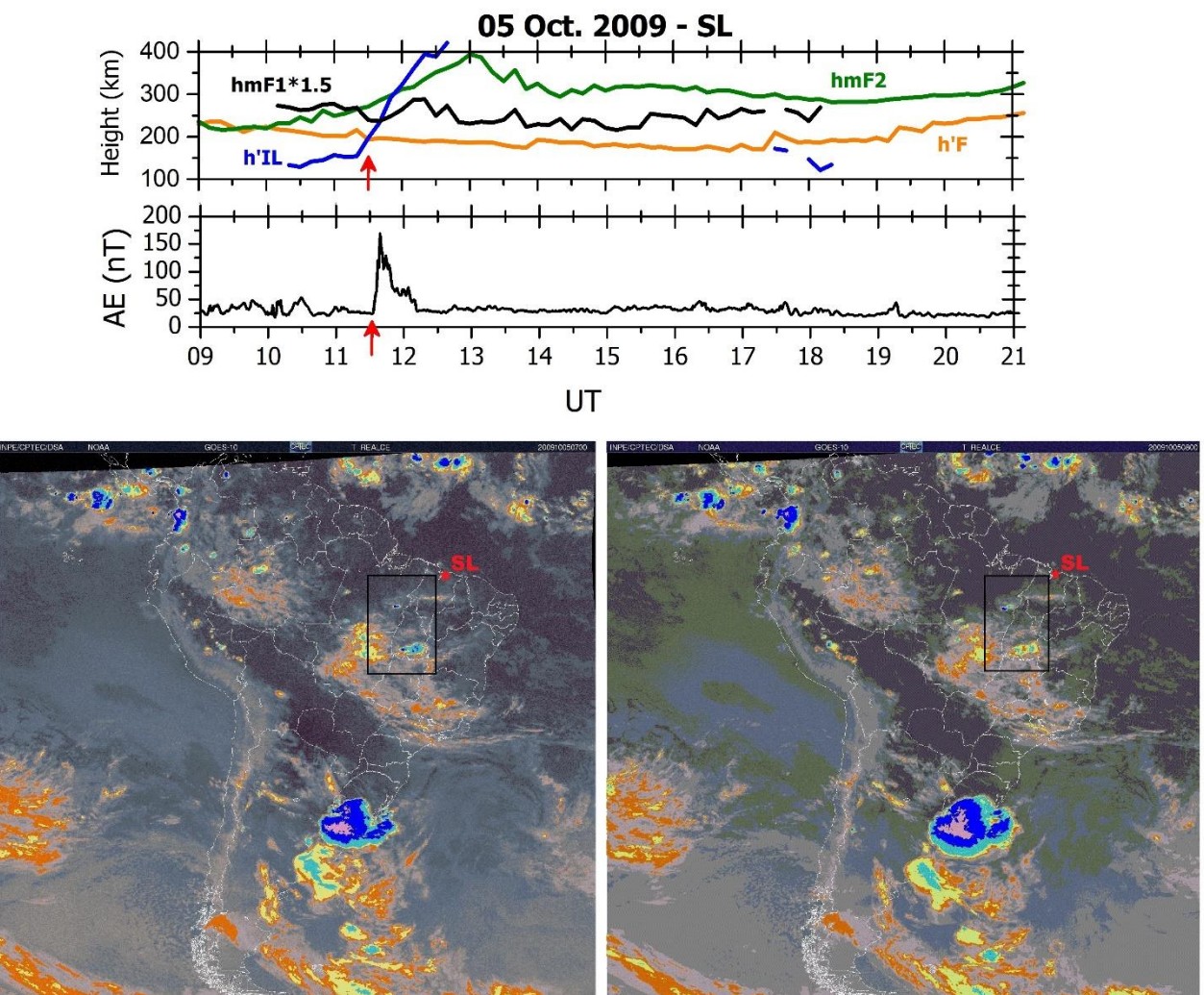

**FIGURE 9** – Ascending intermediate layer over SL during 5 Oct. The first panel (from top to bottom) shows the virtual height of the F layer (h'F, orange line), the F1 and F2 layer peak (hmF1 and hmF2, black and green lines, respectively) and the virtual height of ILs (h'IL, blue line). For a better visualization of the phenomena in discussion, the hmF1 parameter was multiplied by a factor of 1.5. The second panel shows the auroral activity index AE indicating the possible influences of prompt penetration electric field. Images from GOES satellite for 07:00 and 08:00 UT showing the convective zones near São Luis (as indicated by the black rectangle) is shown in the last panel.

The sequence of ionograms in Figure 8 shows that on 05 October, the IL initially detected at ~130 km, attained first the F1 layer and continues its rise until it progressed to the F2 layer base. During this interval, considerable modifications, like a bifurcation of the F2 layer trace (see for example the ionogram from 1210 UT) was observed. As mentioned by Abdu et al. (1982), distortions like this in the F layer trace can be a result of ionospheric disturbances induced by the atmospheric gravity waves. Besides that, our study refers to a period in which the ionosphere was considerably contracted due to the extremely low level of solar fluxes (UV, EUV and X-rays) (Heelis et al., 2009 and Liu et al., 2011). Balan et al. (2012) mentioned that, under these conditions, and considering the absence of severe magnetic disturbances, the tides and waves originating in the lower atmosphere can be expected to register their effects in the thermosphere and ionosphere more easily. Using all-sky image, Essien et al. (2018) showed that the occurrence of small (SSGW) and medium (MSGW) scale gravity waves over São João do Cariri from 2000 to 2010 was higher in 2009 followed by 2008. During 2009, 289 events of SSGW and 66 events of MSGW were identified on 199 nights of observations (175 with a clear-sky and 24 with clouds). These results are very interesting in that they reveal the special conditions of the coupling between the atmospheric layers during this period of deep solar minimum activity.

Figure 10 seems to present a different scenario for São Luís. As indicated by the blue arrows in Figure 10a, an initial perturbation in the F2 layer base was observed at 1540 UT, around 400 km. At 16:30 UT, this disturbance had already reached the F1 layer (~ 350 km). The ionograms show clearly a downward movement of this stratification. At 18:10 UT, the ILs was already located at ~ 170 km. It is also interesting to note the presence of a second intermediate layer (as indicated by purple arrow) that was in progress when a disturbance in the F2 layer was identified. This second layer also presented a

downward movement and oscillated between periods of intensification and weakening. Figure 10b shows in the first panel the height parameters of the layers ( h'F, hmF2 and h'IL). Contour plots from the upper to the bottom panels show, respectively, 1) the variation of plasma frequency as a function of universal time and altitude without the high frequency (periodicities higher than 01:45hours); 2) the reconstructed signal containing only the diurnal (24 hours), semidiurnal (12 hours), terdiurnal (8 hours) and quarter diurnal (6 hours) components; and 3) the difference between the aforementioned profiles (the residual, $\Delta$MHz) highlighting the remaining intrinsic signal's periodicities other than 06:00h $> T >$ 01:45h, in which $T$ is the period. The range of frequency of these panels was limited between 4 and 7 MHz (the methodology applied to extract the variation of plasma frequency as a function of universal time and altitude based on FFT signal reconstruction is described in detail by Brum et al. (2011) and Goncharenko et al. (2013). In the last panel is shown the auroral activity index AE. Through the h'IL pararemeter, we can observe the occurrence of simultaneous intermediate layers on this day. During the interval in which the IL was present, a downward phase propagation in residual values of $\Delta$MHz can be observed, as denoted by the dashed black lines mainly from 17:00 UT and 18:30 UT, thus characterizing the gravity waves signatures. As indicated by the red arrows in the first panel, it is also interesting to note that between 17:30 UT and 17:40 UT there is a discontinuity in the h'F parameter (it increases from ~ 180 to 300 km). This was due to a detachment in the F1 layer base (as can be clearly seen in the ionograms of Figure 10a) that gave origin to an IL starting from 17:40 UT. One hour later, the F1 layer became reestablished itself and at 18:30 UT the IL was connected to the F1 layer base. Similar to what happened on 05 October, in some moments the h'IL was higher than the hmF2. This occurred during the interval in which a perturbation was observed at the lower frequency region of the F2 layer. We believe that in the absence

of any magnetic disturbance, as can be verified by the very weak auroral activity, the main precursor of

the IL on this day is very likely to be the gravity waves.

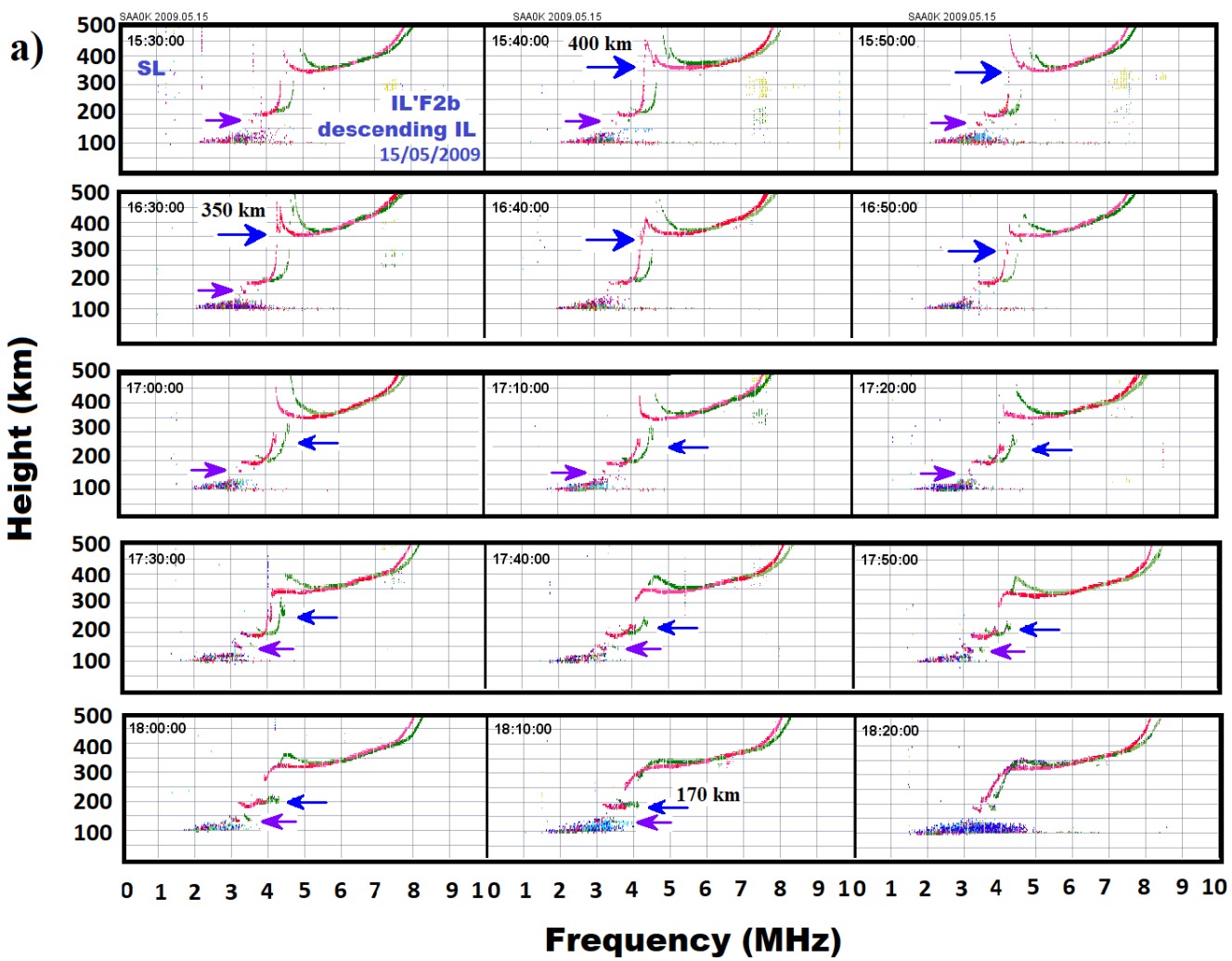

FIGURE 10: a) Sequence of ionograms during 1530 – 1820 UT on 15 May 2009 showing the perturbation in the F2 layer trace that reached low altitudes and evolved to a descending intermediate layer over São Luís. The blue arrow indicates the downward movement of perturbation in F layer and the purple arrow indicates the presence of another IL occurring simultaneously

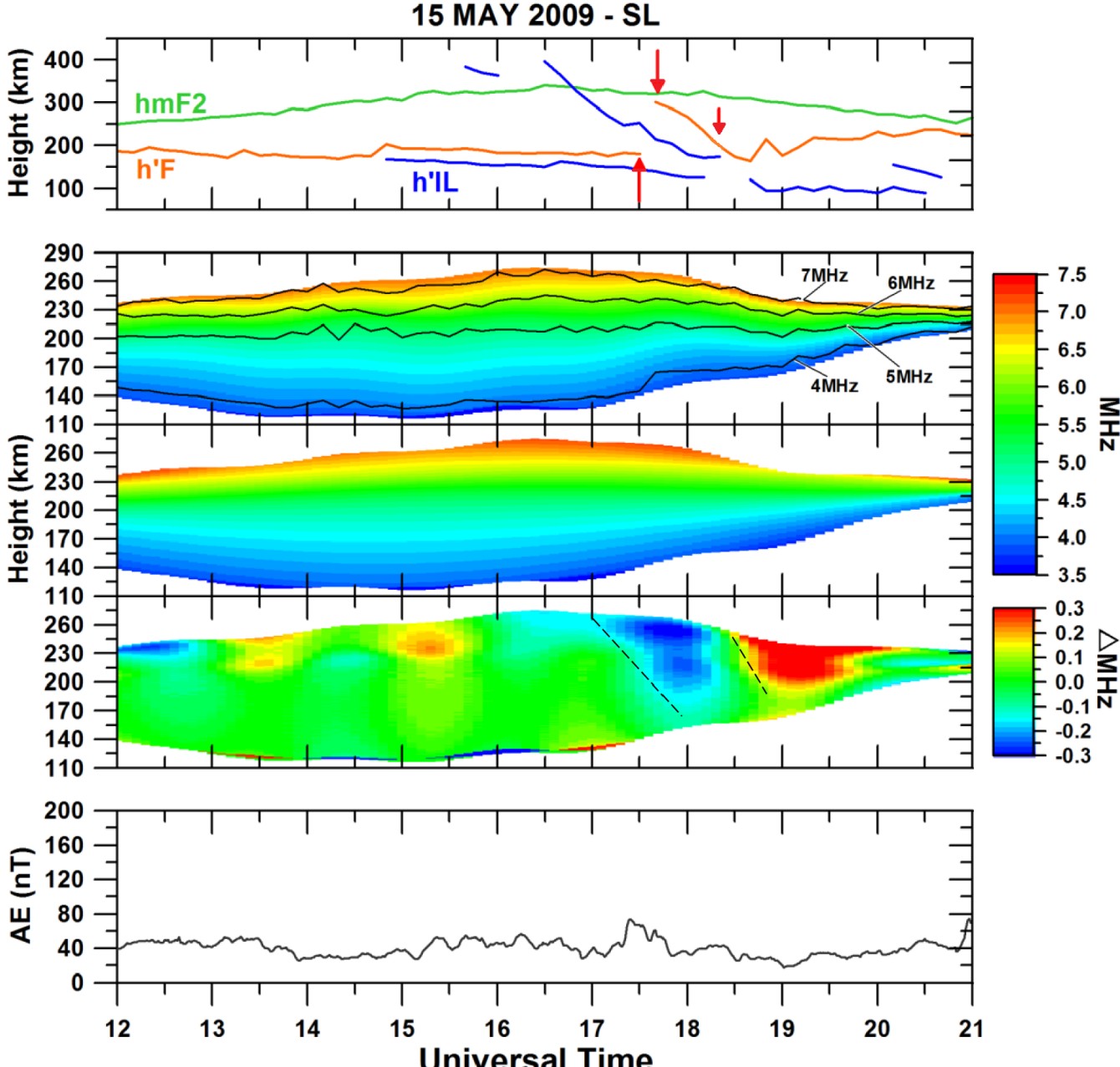

**FIGURE 10**: b) In the first panel are shown the virtual height of the F layer h'F (orange line), the F2 layer peak hmF2 (green line) and the virtual height of ILs (h'IL, blue line). The red arrows indicate some modifications in the F layer in the form of the formation of the IL from a detachment of the F1 layer and the junction of the IL with the F1 layer. From the second to the fourth panel are shown the variation of plasma frequency as a function of universal time and altitude without the high frequency (periodicities higher than 01:45hours) is shown; the reconstructed signal containing only peridiocities

higher than 6 hours and the difference between the aforementioned profiles, respectivelty. The downward phase propagation in the residual ΔMHz is indicated by the dashed gray line. The last panel shows the auroral activity.

5    Figure 11 shows an example of simultaneous intermediate layers over CP. We may note that at 1407 UT there is an IL at ~ 170 km (indicated by the blue arrow) and another at 150 km (indicated by the purple arrow). In both cases, there is a very slow descent movement or nearly a stagnation of the layer at a specific height. The two layers persisted until 17:22 UT (not shown here) after which they merged at ~130 km and continued as a single layer observed until 21:00 UT.

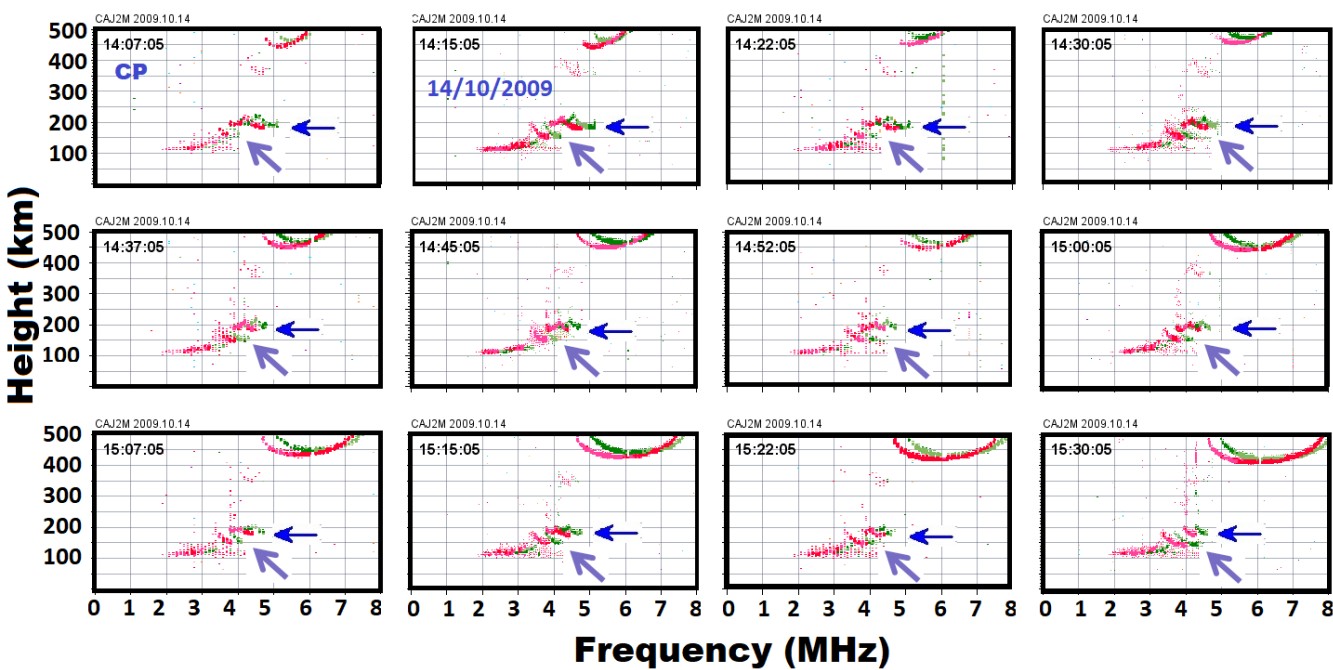

**FIGURE 11:** Sequence ionograms over CP showing the presence of simultaneous intermediate layer, one being initially detected in 180 km height and other in 150 km.

The occurrence of the intermediate layer arising from a detachment of the F1 layer base was the

15    most common case found in our study. The ionograms in Figure 12 show that at 17:15 UT on 18 March,

the F1 layer base exhibited a deformation (stratification), which continued in the following ionograms. This deformation turned into a clear detachment from the F1 layer at 17:30 UT, reaching the height of ~ 150 km at 19:00 UT. It is important to mention that in many other cases, a deformation in the F1 layer base was registered, however a total detachment did not occur or it occurred only later. Besides that, as mentioned previously, there were some cases in which the detachment was verified, but after some time the detached part joined the F1 layer again.

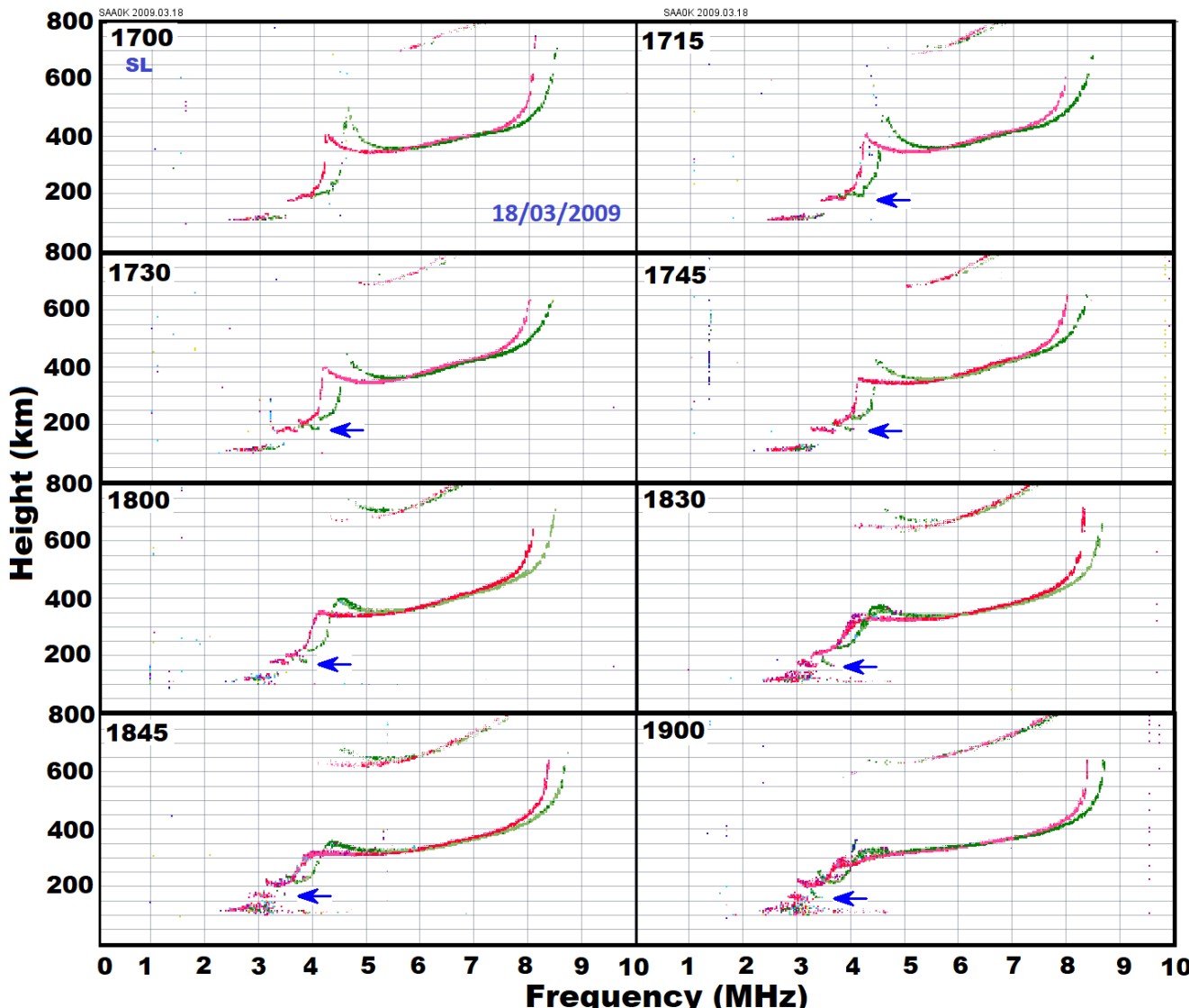

**FIGURE 12**: Sequence of ionograms from SL during 18 March, 2009, showing the formation of an intermediate layer from a detachment of F1 layer base.

The F-layer trace distortion was a frequent feature seen in our data. Figure 13 shows some examples of ionograms over the equatorial station (SL) and the low latitude station (CP) in Brazil. Large modifications in the F1 and F2 layers were noticed, not only as bifurcations as previously mentioned, but

also in the form of forking traces in both F1 and F2 layer, besides other modifications that were not  very well defined. The ionograms from 13 October (SL), 08 December (CP) and 5 October (SL) show some examples of the presence of forking trace. Note that the last example corresponds to the same day for which the case study was presented in Figure 8. On 08 December, the Digisonde registered forking traces over Cachoeria Paulista for more than 2 hours. On 23 July, some interesting aspects also may be noticied. At 17:07 UT, well-defined F1 and F2 layers can be observed. Eight minutes later, the structuring of the F2 layer was considerably modified. The critical frequency decreased from ~ 7.0 to ~5.5 MHz, but the most interesting modifications were verified at 17:30 UT, when the F2 layer detached from the F1 layer and rose by ~ 120 km. Later on, the sequence of ionograms (not presented here) showed that the F1 layer was transformed into a descending intermediate layer, as expected.  A strong uplift of the layer in a short time scale is generally caused by a prompt penetration electric field of eastward polarity (Abdu et al. 2009a; Santos et al. 2012; Santos et al. 2016) but, in this case, the interplanetary magnetic field was weakly to south (~ -2 nT) during its recovering phase. The AE and Dst indices were 150 nT (recovery phase) and -23 nT, respectively. In this scenario, we do not believe that this distortion and uplifting of the layer could be caused by a penetration electric field, because the over-shielding electric field (based on AE recovery), which is westward in this case should make the layer go down and not rise as observed. Therefore, it is highly probable that this modification in the F2 layer and the subsequent formation of the IL was caused by a gravity waves propagation. Some oscillations were also observed over SL (not shown here), with a forking trace observed at 21:00 UT. Adding to that, Figure 13 shows other common examples of distortions occurring in the F1 layer. Oscillation in F layer height like the one present on 18 August, generally evolve into an intermediate layer dettached from the F1 layer base. In another example, on 13

April, a different aspect was noted at the high frequency end of the F1 layer. In the course of time, the characteristic that appeared to be an extra ionization evolved to a descending intermediate layer that attained heights below 130 km, for example.

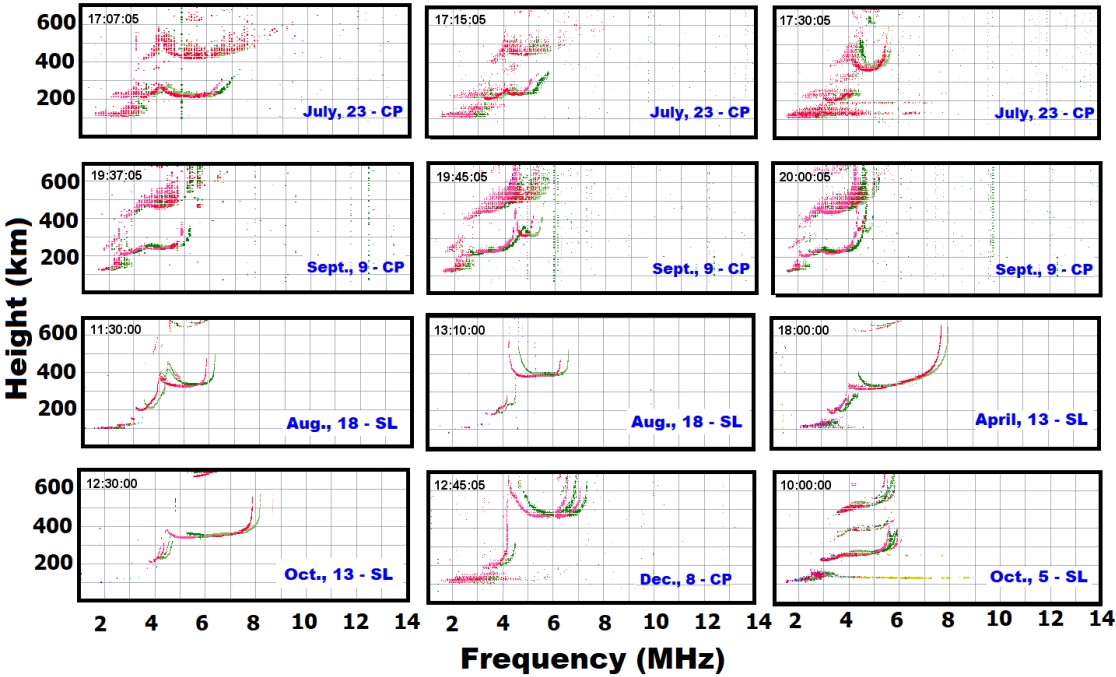

5    **FIGURE 13**: Ionograms showing the possible manifestation of gravity waves in the intermediate layer at equatorial (SL) and low latitude (CP) regions.

## 3.4 Intermediate layer descent velocities

The descent velocities of the ILs for each day is presented in Figure 14. In general, the velocities over São Luis and Cachoeira Paulista vary between -3 m/s and -25 m/s, except on some days in which the velocities attained values lower than -40 m/s. Through the B-spline curve fitting, which is represented by the red curve, we may note a day-to-day variability showing that intermediate layers over both

equatorial and low latitude regions are possibly influenced by a wave-like perturbation with a periodicity of some days. The velocities obtained here are in agreement with those presented by Niranjan et al. (2010). These authors showed a case in which the IL was observed in two different intervals on the same day. In the first interval, the initial descending velocity was around 20 km/h, which decreased to 6 km/h towards the end. In the second interval, the velocity was quite high being 40 km/h in the first 15 minutes and decreasing to 8 km/h before it merged with the normal E layer. The daily velocities plotted in Figure 14 were used to calculate the mean velocity for each month, which is shown in Figure 15. We may note that the mean velocities over CP oscillated between ~ - 13 and -23 km/h, whilst the velocities over SL varied from -18 to -30 km/h. It is interesting to notice that the velocities over SL and CP show similar behavior from March to May. During June and July months, the mean velocities over both sites were very similar, but later clear anti correlation between the velocities can be noted.

a)

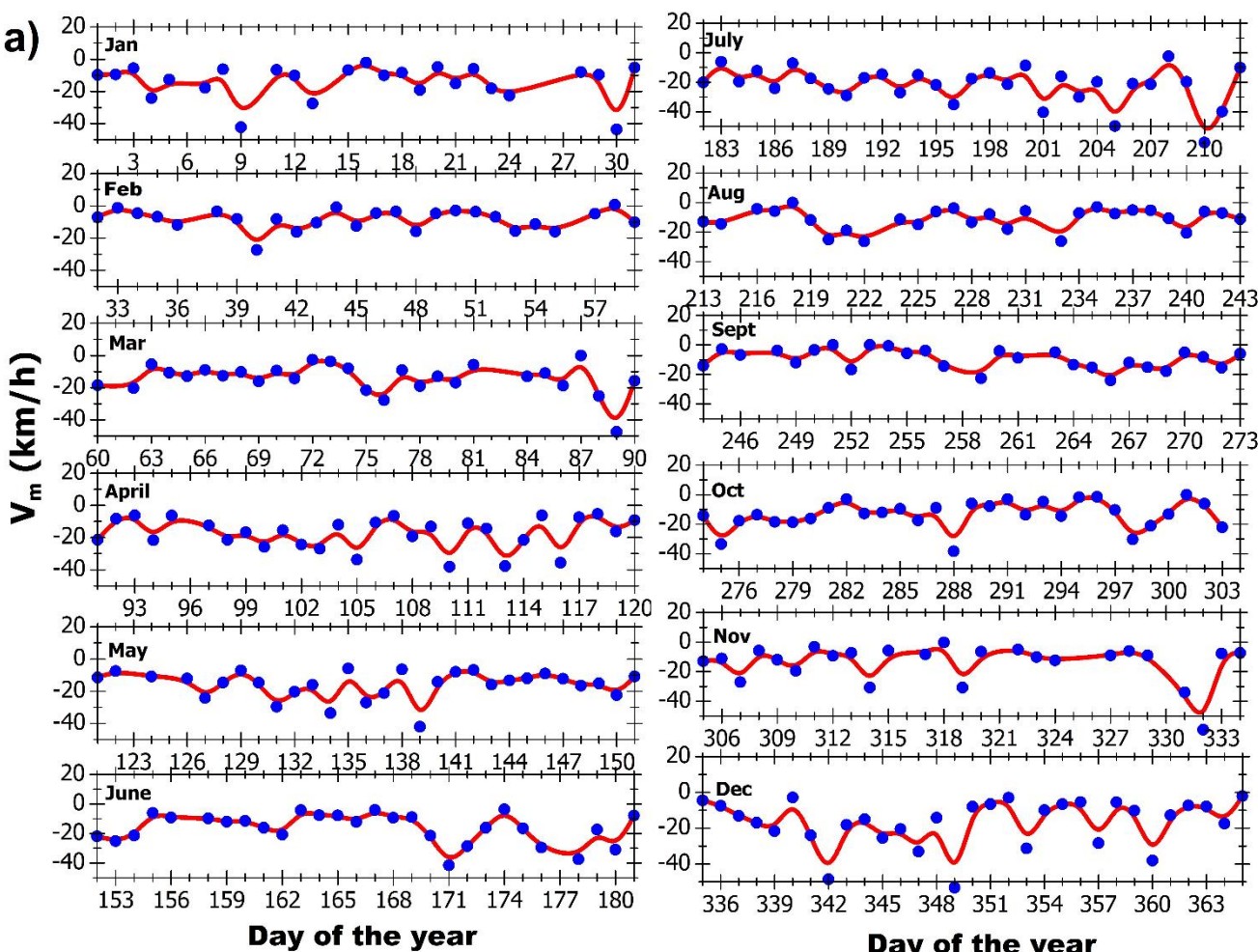

$V_m$ (km/h)

Day of the year

Day of the year

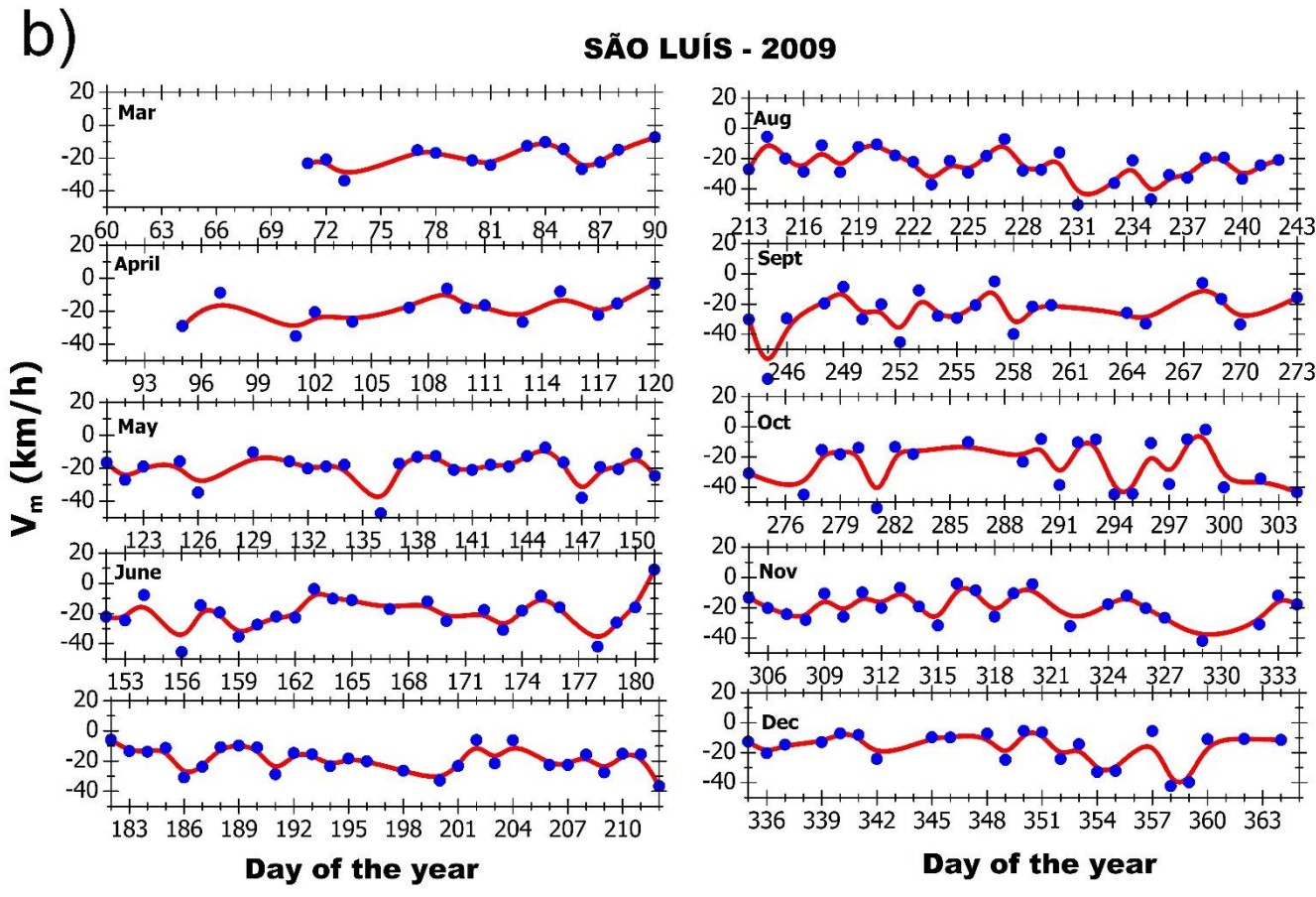

**FIGURE 14:** a) The descending velocity of the intermediate layers for a) Cachoeira Paulista  and b) for São Luís.

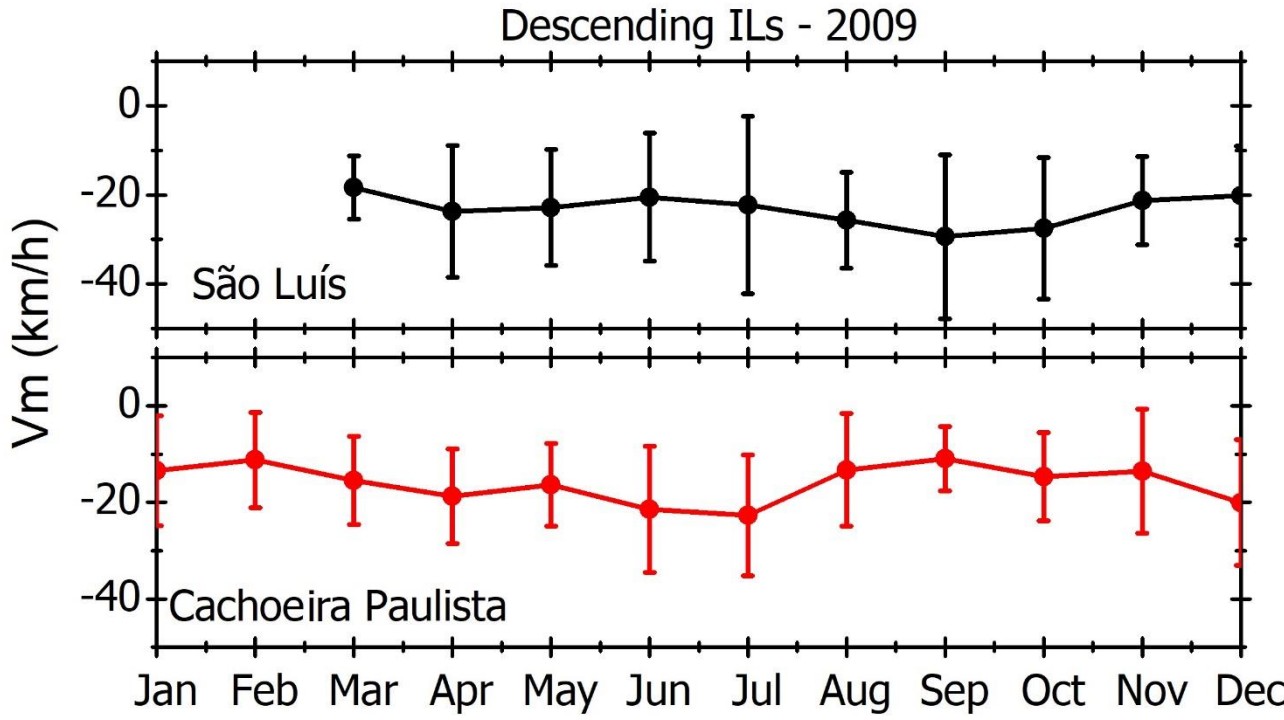

**FIGURE 15:** The monthly average descending velocity of intermediate layers for São Luís and Cachoeira Paulista (black and red profiles, respectively). The vertical bars represent the standard deviation.

## 5. Conclusions

In this work, we have presented for the first time the climatology of the 150 km echoes, or the descending intermediate layers, over the equatorial and low latitudes regions in Brazil during the extended solar minimum of 2009. The occurrence of the descending layers predominantly during daytime (Figure

10    5) appears to suggests an important role played by the E-layer dynamo in the dynamics of these descending layers. The present results would further suggest that this layer has a high probability of occurrence at low latitudes. The top frequency reflected by the IL presents a diurnal variation pattern

similar to that of the normal E layer critical frequency, with a maximum intensity at ~12 LT. The present study appears to reveal that the high occurrence rate of the IL was not related to its seasonality, mainly for CP. Further peculiarities on the layer occurrence were found, such as, the presence of nocturnal layers, simultaneous ILs, ascending ILs and also their formation as a detachment from F1 layer base. It was

shown from Fourier analysis that the dynamics of the ILs over the Brazilian region is dominated by the diurnal tide, followed by the semidiurnal, terdiurnal and quarterdiurnal tides. Additionally, the atmospheric gravity waves propagation and the prompt penetration of magnetospheric electric fields to equatorial latitude may both influence the dynamics of ILs.

*Data availability*. The data used in this study may be acquired by contacting the responsible coordinators at DAE/INPE (Inez S. Batista; e-mail: inez.batista@inpe.br).

*Author contributions.* AMS processed the Digisonde data, performed the analysis and wrote the paper. ISB, MAA, JHAS, JRS and CGMB contributed in the interpretation of the data.

*Competing interests*. The authors declare that they have no conflict of interest.

*Special issue statement.* This article is part of the special issue "7th Brazilian meeting on space geophysics and aeronomy". It is a result of the Brazilian meeting on Space Geophysics and Aeronomy,
Santa Maria/RS, Brazil, 5–9 November 2018.

*Acknowledgments*.  Ângela M. Santos acknowledges the Fundação de Amparo à Pesquisa do Estado de São Paulo – FAPESP for the financial support under grant 2015/25357-4. ISB acknowledges CNPq support under grants 405555/2018-2 and 302920/2014-5. One of us (JHAS) has CNPq grant number

303741/2014-7. J. R. Souza thanks CNPq (grant 307181/2018-9). The auroral indices were obtained from the website https://omniweb.gsfc.nasa.gov/form/omni_min.html. The images from GOES satellite were downloaded from the site: http://satelite.cptec.inpe.br/acervo/goes16.formulario.logic. The Arecibo

Observatory is operated by the University of Central Florida under a cooperative agreement with the National Science Foundation (AST-1744119) and in alliance with Yang Enterprises and Ana G. Méndez-Universidad Metropolitana.

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
