# Peer review of "Climatology of intermediate descending layers (or 150-km echoes) over the equatorial and low latitude regions of Brazil during the deep solar minimum of 2009"

_Annales Geophysicae, 2019_

## Short Comment (SC1) · 21 Jun 2019

Dear Editor, I have reviewed the paper by Santos et al. titled "Climatology of intermediate descending layers (150 km) over the equatorial and low latitude regions of Brazil during the deep solar minimum of 2009". The paper describes digisonde observations of the intermediate layers (IL) over two Brazilian stations: Cachoeira Paulista and San Luis. The IL or the "150 km echo" is of great interest to the aeronomy community, it should be considered for publication. However, I have some issues with the paper in the current form.

1. I would like to have the term '150 km echo' somehow included in the title, which will be useful to many readers who are more familiar with that name. 2. When we get into discussion, we should be talking about things already shown. Yet, we see new things in the discussion. While they may be interesting, they should have been mentioned earlier. 3. There is a discussion about the tidal influence on the IL, but there is no discussion on regional difference and seasonal variations of tides at the two stations. Are the ILs affected by the in-situ tidal force or MLT tides via dynamo effects? There should be more discussion on this topic. Including some tide observation papers will be very useful.

Minor issues 1. What is color code for data in Figure 4? 2. P27, L20, 'some studies have mentioned about this ...' need to add references 3. P3, L5 to L10 'At 17 UT ...' what is the local time? 4. I think the abstract can be shortened to a list of short statements.

---

## Author Comment (AC1) · 17 Jul 2019

Reviewer comments on the paper entitled " Climatology of intermediate descending layers (150 km) over the equatorial and low latitude regions of Brazil during the deep solar minimum of 2009" by Santos et al. and our responses.

Dear Editor, I have reviewed the paper by Santos et al. titled "Climatology of intermediate descending layers (150 km) over the equatorial and low latitude regions of Brazil during the deep solar minimum of 2009". The paper describes digisonde observations

of the intermediate layers (IL) over two Brazilian stations: Cachoeira Paulista and San Luis. The IL or the "150 km echo" is of great interest to the aeronomy community, it should be considered for publication. However, I have some issues with the paper in the current form. 1. I would like to have the term '150 km echo' somehow included in the title, which will be useful to many readers who are more familiar with that name. 2. When we get into discussion, we should be talking about things already shown. Yet, we see new things in the discussion. While they may be interesting, they should have been mentioned earlier. 3. There is a discussion about the tidal influence on the IL, but there is no discussion on regional difference and seasonal variations of tides at the two stations. Are the ILs affected by the in-situ tidal force or MLT tides via dynamo effects? There should be more discussion on this topic. Including some tide observation papers will be very useful. Minor issues 1. What is color code for data in Figure 4? 2. P27, L20, 'some studies have mentioned about this ...' need to add references 3. P3, L5 to L10 'At 17 UT ...' what is the local time? 4. I think the abstract can be shortened to a list of short statements.

We thank the reviewer for careful reading of our manuscript and finding it worthy for publication after minor revision. Our responses to the specific comments are given below:

1. I would like to have the term '150 km echo' somehow included in the title, which will be useful to many readers who are more familiar with that name. The term "150-km echo" was included in the title as suggested by the reviewer. The new title is: "Climatology of intermediate descending layers (or 150-km echoes) over the equatorial and low latitude regions of Brazil during the deep solar minimum of 2009".

2. When we get into discussion, we should be talking about things already shown. Yet, we see new things in the discussion. While they may be interesting, they should have been mentioned earlier. We thank the referee for this suggestion. We create a new section in which the results and discussions are presented together. See P5, L5.

3. There is a discussion about the tidal inïñĆuence on the IL, but there is no discussion on regional difference and seasonal variations of tides at the two stations. Are the ILs affected by the in-situ tidal force or MLT tides via dynamo effects? There should be more discussion on this topic. Including some tide observation papers will be very useful. A discussion about the variability of atmospheric tide with the latitude and seasonality was included in the manuscript in the P17, L1 to L12 and some references were added as suggested by the referee. With base in our data analysis, we believe that the IL's formation is primary control by driven meridional and zonal wind shear forces and that the day-to-day variability can be due the variations in tides, gravity waves, electric fields and metallic ions population.

Minor issues

1. What is color code for data in Figure 4?

Different colors in the panels of Figure 4a and 4b were used to represent the days of each month (P10 to P11).

2. P27, L20, 'some studies have mentioned about this ...' need to add references References were added as suggested. See P22, L6 to L8.

3. P3, L5 to L10 'At 17 UT ...' what is the local time? The local time was included in this part of the text (see P3, L6). The local time over the Brazilian sector is given by LT = UT − 3h. This information was included in P3, L25.

4. I think the abstract can be shortened to a list of short statements. Some changes have been done in the abstract as to meet the referee's suggestion, see P1.

Please also note the supplement to this comment:
https://www.ann-geophys-discuss.net/angeo-2019-74/angeo-2019-74-AC1-supplement.pdf

2019.

**Supplement:**

[revised manuscript text omitted]

---

## Referee Comment (RC2) · Anonymous Referee #2 · 19 Jul 2019

Dear Editor,

The paper titled "Climatology of intermediate descending layers (150 km) over the equatorial and low latitude regions of Brazil during the deep solar minimum of 2009" presents a statistical study of intermediate layers (IL's) using digisondes observations. The statistical results are new, interesting and can be helpful to understand the formation and dynamics of the IL's. Therefore, the paper should be considered for publication after revision. This way, I have listed below some comments/questions/suggestions.

[Figure]

1. P20, L1-3: I cannot see a relation between the day-to-day variability shown in Figures 10a and 10b and gravity waves. Figures 10a and 10b show a wave like perturbation with a periodicity of some days. Gravity waves periodicity varies from some minutes to few hours. It is not possible to affirm the influence of gravity waves with one point per day. This discussion needs some improvements.

2. P29, Figure 14 and its discussion: a) In the label of Figure 14, h'F line is yellow and h'IL line is blue; b) How can h'IL be higher than hmF2? c) hmF2 is not intensified at 1115 UT, it has a smooth upward movement starting at around 0930 UT. Apparently, not related with h'IL intensification; d) It is not possible to observe gravity wave activity in the F layer true height at fixed plasma frequencies. Characteristics of gravity waves as downward phase propagation cannot be seen, only a modulation that could be related to prompt penetration electric field. Would be helpful if the authors presented others plasma frequencies, e. g., 5, 6, 7, 8 MHz, in order to see a vertical propagation of gravity waves in the ionosphere. 4.1, 4.2, 4.3, 4.4 MHz represent basically the same ionospheric height, which makes difficult to see any gravity wave propagation; e) It is also difficult to address a possible cause to gravity wave when we have a magnetic disturbance. The author should choose a case without any magnetic disturbance and try to improve this discussion, even the authors do not believing that the uplifting of the layer was caused by a penetration of electric field.

3. a) P1, L12: "São Luís (2 ° S; 44 ° W, I: -5.7°)" should be "São Luís (2° S; 44° W, I: -5.7°)"; b) P3, Methodology and data presentation: Some details about the digisonde used could be helpful for those who don't know about it (e. g., model, time resolution, etc) or, at least, some references; c) P5, Figure 1: Would be nice know the time of occurrence of the ionogram, even in an example; d) P5, L10: What "60%" means? 60% of the days analyzed for each month or 60% of the ionograms? General information about the statistic (e.g., number of day with data and number of day with IL's), as did in Table 1, could be summarized in this section; e) P6, Figure 2: What kind of mean have done in Figure 2? Does it actually a monthly mean? f) P8, Figure 3: Standardize (0 to

24h) both x-axis (São Luis and Cachoeira); g) P8, L6: It is not possible to distinguish the days in this plot. We cannot check the number of days with or without IL's; h) P9, L1: Why have you not seen IL's in March, April, and September (very small occurrence) around 15 UT? i) P20, L11-12: "SL/CP" should be "CP/SL"; j) P24, Figure 12: The number of IL's doesn't match with the information given in Table 1.

———————————————————

---

## Referee Comment (RC3) · Anonymous Referee #1 · 22 Jul 2019

Dear Editor, I am happy with the revisions.

---

## Author Comment (AC2) · 3 Sep 2019

Response letter to Reviewer #2

Reviewer comments on the paper entitled " Climatology of intermediate descending layers (150 km) over the equatorial and low latitude regions of Brazil during the deep solar minimum of 2009" by Santos et al. and our responses.

Dear Editor, The paper titled "Climatology of intermediate descending layers (150 km) over the equatorial and low latitude regions of Brazil during the deep solar min-

imum of 2009" presents a statistical study of intermediate layers (ILs) using digison-des observations. The statistical results are new, interesting and can be helpful to understand the formation and dynamics of the ILs. Therefore, the paper should be considered for publication after revision. This way, I have listed below some com-ments/questions/suggestions.

Our answers: We thank the reviewer for taking the time to review our manuscript and provide important comments and questions. Our responses are given below:

Discussion paper 1. P20, L1-3: I cannot see a relation between the day-to-day vari-ability shown in Figures 10a and 10b and gravity waves. Figures 10a and 10b show a wave like perturbation with a periodicity of some days. Gravity waves periodicity varies from some minutes to few hours. It is not possible to affirm the influence of gravity waves with one point per day. This discussion needs some improvements. We thank the reviewer for bringing out this important point. We agree with the reviewer that it is not possible to verify the gravity waves influence with one point per day. We corrected this part of the manuscript. See P34, L9-12 and P35, L1-2 (also please note that the illustrations numbering have changed in the revised version).

2. P29, Figure 14 and its discussion: a) In the label of Figure 14, h'F line is yellow and h'IL line is blue; Ok, corrected (please note that the illustrations numbering have changed in the revised version).

b) How can h'IL be higher than hmF2? As showed in the ionograms of Figure 8, the ionosphere behavior was very peculiar during this day (05 October). It is possible to check in the ionogram at 1020 UT, the IL was located at ∼133 km. Until 1120 UT, the variation of the IL height was very clear in the ionograms, but in the next times, a complex behavior was observed. As discussed in P24, L5-14, the IL appeared to have merged with the F1 layer at 1130 UT. After this junction, the h'IL was considered based on the perturbation of the extra ionization at the high frequency end of the F1 layer until this perturbation attained the F2 layer. The virtual height (h'IL) at a given frequency is

higher than the true height at the same frequency. In the present case, from 12 UT the h'IL was higher than the hmF2 as the reviewer can noted. This can be easily verified in the ionograms show below in which the grey curve represents the true height derived from the ionogram:

c) hmF2 is not intensified at 1115UT,it has a smooth upward movement starting at around 0930 UT. Apparently, not related with h'IL intensification; Yes, the variations in the hmF2 were very smooth when compared with the variations in IL. We believe that a PPEF can have influenced the IL movement during this event, however the reason for the distinct responses of the ionospheric parameters to the penetration electric field is not completed understood and would need more investigation. An explanation on this matter was included in the P25, L1-5.

d) It is not possible to observe gravity wave activity in the F layer true height at fixed plasma frequencies. Characteristics of gravity waves as downward phase propagation cannot be seen, only a modulation that could be related to prompt penetration electric field. Would be helpful if the authors presented others plasma frequencies, e. g., 5, 6, 7, 8 MHz, in order to see a vertical propagation of gravity waves in the ionosphere. 4.1, 4.2, 4.3, 4.4 MHz represent basically the same ionospheric height, which makes difficult to see any gravity wave propagation; We prepared a new figure following the suggestion of the referee using the plasma frequencies of 3, 4 and 5 MHz. It was not possible to analyze the 6 and 7 MHz frequencies due to data gaps in the interval of interest. Now a downward phase propagation can be noted in Fig. 9 (P26). Although not very clear, we also can observe in the first peak oscillation, between 10 and 1130 UT, some kinks (mainly in the 3 and 4 MHz), that can indicate some perturbation caused by a gravity wave. A discussion about this new figure was included in P24, L5-14.

e) It is also difficult to address a possible cause to gravity wave when we have a magnetic disturbance. The author should choose a case without any magnetic disturbance and try to improve this discussion, even the authors do not believing that the

uplifting of the layer was caused by a penetration of electric field. We agree with the reviewer. We include now a new case study (Figure 10b, P30) in which we believe that the magnetic disturbance does not affect the behavior of the intermediate layers. See the discussion about this in P28, L1-15.

3. a) P1, L12: "São Luís (2 ◦ S; 44 ◦ W, I: -5.7◦)" should be "São Luís (2◦ S; 44◦ W, I: -5.7◦)"; Ok. Done.

b) P3, Methodology and data presentation: Some details about the digisonde used could be helpful for those who don't know about it (e. g., model, time resolution, etc) or, at least, some references; A brief introduction about the Digisonde was included in P4, L1-10.

c) P5, Figure 1: Would be nice know the time of occurrence of the ionogram, even in an example; The ionogram in Figure 1 was registered at 1630 UT. This new information is included in the Figure. See P5.

d) P5, L10: What"60%"means? 60% of the days analyzed for each month or 60% of the ionograms? General information about the statistic (e.g., number of day with data and number of day with ILs), as did in Table 1, could be summarized in this section; The percentage of occurrence of the intermediate layers was calculated considering the number of days with intermediate layers and the number of days of available data. For example: in March (SL) the calculation considered 21 days with available data, being 15 of them with the presence of IL. This give a percentage of occurrence of $\sim$ 70%. A table containing the number of data used in the calculation for all months was included in the manuscript as suggested by the reviewer. See P6.

e) P6, Figure 2: What kind of mean have done in Figure 2? Does it actually a monthly mean? As explained in previous item (d), the data presented in Figure 2 correspond to the monthly percentage occurrence of the ILs and not a "monthly mean percentage occurrence" as we wrote in the legend of the Figure 2. The word "mean" was inappropriately used and was removed.

[Figure]

f) P8, Figure 3: Standardize (0 to 24h) both x-axis (São Luis and Cachoeira); The x-axis was standardized.

g) P8, L6: It is not possible to distinguish the days in this plot. We cannot check the number of days with or without ILs; The reviewer is right. This part was removed from the manuscript.

h) P9, L1: Why have you not seen ILs in March, April, and September (very small occurrence) around 15 UT? This ILs behavior was very interesting and occurred only over SL. In March, the lack of data (10 days) may have influenced this result, but in the other months (April and September) the number of days without data was very small or none during this time. In these months, we verify that during the interval between 12 and 15 UT, a perturbation in the F1 layer was present, but for some reason, the IL formation from a detachment in the F1 layer base (as we observe in many other cases) did not occur, or occurred only later on. A detailed investigation about this point needs to be done, but in this moment, this analysis is out of scope in this work.

i) P20, L11-12: "SL/CP" should be "CP/SL"; After the inclusion of the standard deviation in Figure 15, we think better remove this text of the manuscript.

j) P24, Figure 12: The number of ILs doesn't match with the information given in Table 1. The number of ILs does not match with the information given in former Table 1, because in the case of Figure 12 (Figure 5 in the new version of the manuscript), all the simultaneous ILs observed were considered in the calculation of the occurrence probability. In Figure 2, we consider only the occurrence or non-occurrence of the IL at each day, independently if the ILs were observed one or more times during a given day. In the equinox (March-April, September – October), for example, the occurrence probability of the ILs over SL in Figure 2 (using the data from the new Table 1) was calculated considering 85 days with the presence of IL. In the calculation of Figure 5 (former Figure 12), we considered 86 IL events, because on 17 April, we observed the presence of two ILs occurring at the same time (two events in the same day). An

explanation of this was included in P13, L14-18.

Please also note the supplement to this comment:
https://www.ann-geophys-discuss.net/angeo-2019-74/angeo-2019-74-AC2-supplement.pdf

**Supplement:**

[revised manuscript text omitted]

---

## Author Comment (AC3) · 5 Sep 2019

We thank reviewer 1 for accepting our corrections.
* * *

---

## Author Response (AR1)

**RESPONSE LETTER TO EDITOR**

*Dear Angela Santos et al.*

*Thank you for revise the manuscript ANGEO-2019-74. It is clear that your manuscript has been improved after the revision. I would like to thank the both reviewers for important suggestions/comments that they have posted. Based on your responses and the new version of the manuscript, I still not convinced about the presence of gravity waves in figures 9 and 10b). I guess the filtering process that you are using is not good to show the phase propagation of gravity waves in the dhF. I mean, the phase structures were not well defined and they have changed from a fixed frequency line to another. May you, please, work to improve these figures? I am going to consider the paper for publication in the Annales Geophysicae after your feedback.*

*Best regards,*

*Igo*

**Our answer:**

We thank the editor for considering our paper for publication after minor revisions. We believe that the mixed effects from disturbed electric fields and gravity waves can make it difficult for the visualization of the downward phase propagation in dhF parameter. Thus, we think it was better to remove this specific analysis from this plot (second panel from top to bottom in the former Figure 9). See the new Figure 9 in Page 25 and the new explanation about it in Page 24, lines 14 to 16. Regarding Figure 10a, in which the ILs occurred in the absence of the magnetic storm events, a new data filtering and processing methodology were used enhancing the phenomena signature that we would like to highlight and now the downward phase propagation of the gravity waves is more clearly seen. The description of this Figure can be found on Pages 27-28 and the new Figure 10b on Page 29.

Dear Editor, please consider the inclusion of Dr. Christiano Brum as co-author of this paper.

Thank you,

Ângela Santos

---

## Author Response (AR2)

*Dear Dr. Santos!*

*Thank you upload for the revised version of the manuscript. As all concerns from the editor and reviewers were properly addressed, I recommend to publish the manuscript as it is. If possible, before to submit the final version of the manuscript, please, write the contribution of all authors.*

5 *Best regards,*

*Igo*

**Dear Editor**

**The information about the author contributions, as data availability and competing interests were** 10 **included in the manuscript.**

**Ângela**